# Resilience of UK crop yields to compound climate change

Louise J. Slater[1*], Chris Huntingford[2], Richard F. Pywell[2], John W. Redhead[2], Elizabeth J. Kendon[3,4]

[1]School of Geography and the Environment, University of Oxford, Oxford, OX1 3QY, UK
[2]UK Centre for Ecology and Hydrology, Wallingford, Oxon, OX10 8BB, UK
[3]Met Office, FitzRoy Road, Exeter, Devon, EX1 3PB, UK
[4]Bristol University, Faculty of Science, BS8 1UH, UK

*Correspondence to*: Louise J. Slater (louise.slater@ouce.ox.ac.uk)

**Abstract.** Recent extreme weather events have had severe impacts on UK crop yields, and so there is concern that a greater frequency of extremes could affect crop production in a changing climate. Here we investigate the impacts of future climate change on wheat, the most widely grown cereal crop globally, in a temperate country with currently favourable wheat-growing conditions. Historically, following the plateau of UK wheat yields since the 1990s, we find there has been a recent significant increase in wheat yield volatility, which is only partially explained by seasonal metrics of temperature and precipitation across key wheat growth stages (Foundation, Construction and Production). We find climate impacts on wheat yields are strongest in years with compound weather extremes across multiple growth stages (e.g. frost and heavy rainfall). To assess how these conditions might evolve in the future, we analyse the latest 2.2km UK Climate Projections (UKCP Local): on average, the Foundation growth stage (broadly 1st October to 9th April) is likely to become warmer and wetter, while the Construction (10th April to 10th June) and Production (11th June to 26th July) stages are likely to become warmer and slightly drier. Statistical wheat yield projections, obtained by driving the regression model with UKCP Local simulations of precipitation and temperature for the UK's three main wheat-growing regions, indicate continued growth of crop yields in the coming decades. Significantly warmer projected winter night temperatures offset the negative impacts of increasing rainfall during the Foundation stage, while warmer day temperatures and drier conditions are generally beneficial to yields in the Production stage. This work suggests that on average, at the regional scale, climate change is likely to have more positive impacts on UK wheat yields than previously considered. Against this background of positive change, however, our work illustrates that wheat farming in the UK is likely to move outside of the climatic envelope that it has previously experienced, increasing the risk of unseen weather conditions such as intense local thunderstorms or prolonged droughts, which are beyond the scope of this paper.

**Short summary**. This work considers how wheat yields are affected by weather conditions during the three main wheat growth stages in the UK. Impacts are strongest in years with compound weather extremes across multiple growth stages. Future climate projections are beneficial for wheat yields, on average, but indicate a high risk of unseen weather conditions which farmers may struggle to adapt to and mitigate against.

# 1 Introduction

Globally, wheat is the most widely grown cereal crop by area, with more than 214 million hectares harvested and an annual production of about 730 million tonnes (FAO, 2018). In the UK, wheat is the most prevalent arable crop, with an annual planting of approximately 1.7 million hectares (ha) (DEFRA, 2018a). The UK climate has historically been well suited to growing wheat (Reynolds, 2010), partly due to technology and investment in the agricultural sector (as can be seen from the increasing trend in **Figure 1a** as technological and agronomic innovations were introduced), but also due to the UK climate, which is suitable to temperate species when autumn-sown (Harkness et al., 2020; Reynolds, 2010). UK yields are of approximately 8 t ha$^{-1}$ (**Figure 1a-b**) compared to a global average of 3.5 t ha$^{-1}$ (FAO, 2018). However, recent climate extremes such as the UK hot summer of 2018 and wet autumn of 2019 had substantial negative impacts on farm businesses, with significant reductions in crop yields. This climate-mediated reduction of yields is supported by evidence from the UK government (DEFRA, 2018b, 2019), the farming industry (AHDB, 2020) and real-time precision yield monitoring (Hunt et al., 2019).

Observed, direct impacts of climate change on crop yields are emerging globally (Brisson et al., 2010; Grassini et al., 2013; Hochman et al., 2017; Rigden et al., 2020), slowing the growth in global agricultural productivity (Ortiz-Bobea et al., 2021) and altering patterns of global food production (Ray et al., 2019). Rising temperatures under anthropogenic climate change are often detrimental to agricultural productivity (Ortiz-Bobea et al., 2021) and compound heat-drought impacts may directly affect crop growth: for instance, maize and soil yields are historically worse in places with strong associations between low rainfall and high temperature (Lesk et al., 2021). Cool and wet growth phases have also been linked to poor yields, because it is hard to warm the surface when soils are wet, and hard to dry wet soils during cooler periods. Thus it remains to be seen how warming and precipitation interact, and whether future warming may help offset the increased precipitation by drying out waterlogged soils. This interaction depends on how the link between precipitation and soil moisture may evolve in the future (a topic drawing increasing attention in both climate and crop science, enabled by the rise of satellite-derived soil moisture observations). Combined with the nutrition demands of a rapidly growing global population, there is an urgent requirement to estimate these effects on future crop yields. Breeding and evaluating new wheat varieties tolerant of hotter, drier summers may take decades (Zheng et al., 2012), and it is unclear whether advances in agronomy are occurring fast enough to mitigate the impacts of any accelerating frequency of extreme climatic events (Chen, D. et al., 2021). Changing climatic conditions may also affect yields indirectly by constraining the ability of farmers to undertake key management actions of tillage, sowing and harvest, or by causing damage to natural capital, such as soil erosion. These new constraints on yields may overtake any gains from physiological and phenological advances obtained through plant breeding.

In order to assess this risk to future food production, there is a critical need to understand how climate extremes are likely to evolve during the seasonal growth phases that are most relevant to the farming industry. Observational evidence has revealed changes in the intensity, frequency, duration, and extent of weather extremes, such as heavy rainfall events and hot days, across certain regions and continents (Rahmstorf and Coumou, 2011; Slater et al., 2021). There has been much research

relating weather indices to potential crop variability or projected damage (Harkness et al., 2020; Iizumi and Ramankutty, 2016; Rosenzweig et al., 2001; Trnka et al., 2014), but most work has described weather extremes by using seasonal or annual metrics rather than focussing on the periods most relevant to crop growth (Frich et al., 2002; Zhang et al., 2011). There is also increasing research focus on compound weather extremes (Zscheischler et al., 2020) occurring simultaneously or in close succession, such as very warm temperatures in the late autumn followed by abnormally wet conditions in spring (Ben-Ari et

al., 2018) and their impacts on crop yields. Of the total annual crop losses in world agriculture, many are due to direct weather and climatic effects such as drought, flash floods, heavy rainfall in otherwise dry periods, frost, hail, and storms (Ray et al., 2019; Sultan et al., 2019). High temperatures and heat stress lead to stomatal closure and therefore reduced photosynthesis due to restricted $CO_2$ diffusion (Chaves et al., 2003), offsetting potential yield gains that might otherwise occur with greater fertilization in a $CO_2$-enriched environment (Ainsworth and Long, 2021). In some regions of the mid and high

latitudes, water excess may prove more detrimental to wheat yields than drought (Zampieri et al., 2017). However, for crops such as maize and soy yields, it has also been shown that heavy rainfall of up to 20 mm hr$^{-1}$ may even prove beneficial, highlighting the benefits of rainfall intensification in a warming climate (Lesk et al., 2020). Overall, there is thus a need to investigate historical data to elucidate the linkage between extreme temperature and rainfall over the agricultural phases of relevance to crop growth. Climate models may then be employed to explore how such linkages might evolve as the climate

warms.

This work thus investigates: (1) whether statistically significant associations exist between observed temperature/precipitation metrics and historical wheat yields during the three crop growth stages, in the three main wheat-growing regions of the UK; and (2) the extent to which projections of compound temperature and precipitation extremes under a high-emissions scenario may impact future crop yields. To assess future changes in precipitation and temperature extremes, we employ state-of-the-art

UK Climate Projections Local (UKCP 2.2km) convection-permitting simulations, which constitute a step-change in resolving small-scale processes in the atmosphere. These climate projections are considered the most reliable simulations presently available in terms of their ability to project future changes in meteorological extremes over the UK.

## 2 Methods

### 2.1 Wheat yield data

Geographically, we focus on the three main wheat-growing regions outlined using the EU "NUTS" classification (European Commission, 2010). These three regions are (i) North Eastern Scotland, Eastern Scotland, and the North East English region (SNE); (ii) East Midlands, Yorkshire and the Humber regions (EMYH); and (iii) South East and Eastern region (SEE) (**Figure 1c-d**). These three regions account for over 80% of total UK wheat production by tonnage (DEFRA, 2015) and correspond with the yield reporting boundaries of available data. The regional wheat yield data were obtained from the UK Department

for Environment, Food and Rural Affairs (Defra) (DEFRA, 2015). The data are drawn from the England Cereals and Oilseeds

Production Survey and Scotland Cereal Production and Disposal Survey, part of an annual survey of the UK agricultural industry. For full details of the survey methodology, see (DEFRA, 2018b). The data were summarised by Defra to average yield at the national (1885-2020) and regional (1990-2020) levels, resulting in 136 and 31 years of data, respectively.

The dates for the Foundation, Construction, and Production growth stages are taken from benchmarks in the UK's 'Wheat growth guide', in **Table 1** (AHDB, 2018). Prior knowledge on the effects of climate in different growth stages guides our choice of climate variables in the study (**Table 1**). Absolute anomalies of wheat yields were computed by fitting a locally-weighted scatterplot smoothing curve (LOESS) to obtain the running mean (red lines shown in **Figure 1a-b**), and subtracting this running mean from each annual value (resulting anomalies shown in **Figure 1c**). We perform this calculation to remove the trend and thereby isolate annual anomalies, which we expect to be related to inter-annual climate variability rather than other factors such as long-term technological improvements, increasing atmospheric carbon dioxide, or climate warming.

## 2.2 Historical precipitation and temperature reference data

For historical climate data we employ the HadUK gridded 5km observational data from the National Climate Information Centre (NCIC) (Hollis et al., 2019). Provisional HadUK data were employed for the year 2020, produced as per previous years (Hollis et al., 2019); provisional data may have very small differences at regional scales compared with the final published dataset, available later in the year. Observed precipitation and temperature data were checked for completeness: any incomplete climate data during each of the crop growth stages (i.e. a Foundation phase with less than 187 days of data; a Construction phase with less than 60 days, or a Production phase with less than 46 days) were removed, to ensure consistency and comparability across years.

To investigate the association with crop yields, we computed climate metrics within each geographical region and wheat growth stage (**Table 2**) using region-averaged values of temperature (°C) and precipitation (mm). Specifically, for temperature, we derived the maximum, mean, and minimum of the region-averaged maximum daily temperature (*max_maxT*, *mean_maxT*, *min_maxT*), of the mean daily temperature (*max_meanT*, *mean_meanT*, *min_meanT*), and of the minimum daily temperature (*max_minT*, *mean_minT*, *min_minT*). For example, *max_maxT* indicates the day with the hottest (maximum hourly) temperature, and *max_minT* indicates the day with the warmest night-time (minimum hourly) temperature, during a given growth stage. We also create metrics representing the daily variability of temperature (*var_dailyT*) and its seasonal variability (*var_maxT*, *var_meanT*, *var_minT*). For instance, *var_maxT* indicates the difference between the highest/lowest daily values of maximum hourly temperature in a season.

For precipitation, we computed metrics representing the total region-averaged daily precipitation within a growth stage (*total_P*) and its quantiles (*max_dailyP* or *mean_dailyP*), where *max_dailyP* is the maximum total daily precipitation within a growth stage. We also considered the variability of daily precipitation across a growth stage (*varP_Q0.95-Q0.05*); the number of heavy rainfall days where precipitation exceeds 10mm (*days_P>10mm*); and the number of dry days where precipitation is less than 0.01mm (*days_P<0.01mm*) (**Table 2**). The heavy precipitation threshold is chosen based on the historical wheat yield

literature for the British Isles (Thomas et al., 1989); other thresholds may be more relevant elsewhere. For instance, (Lesk et al., 2020) found extreme rainfall impacts only at especially high intensities >50mm/hour for US maize and soy; others have used more holistic distributional measures like the wet-day Gini coefficient (Shortridge, 2019), a measure of daily rainfall variability.

**2.3 UKCP Local (2.2km) projections**

The UKCP Local simulations have a spatial resolution of just 2.2km – providing exceptional detail in local rainfall changes. Importantly, such high resolution allows the climate model to explicitly represent convective precipitation events on the model grid (see (Kendon et al., 2019, 2020) for details), thus providing credible projections of future changes in short-duration precipitation extremes, and in particular for summer months. The UKCP Local simulations were initially released in September 2019 (Kendon et al., 2019) but were then updated in July 2021 after correction of an error in the representation of graupel (soft ice pellets) (Kendon et al., 2021). Here we use the new updated Local 2.2km projections. The local 2.2km model (HadREM3-RA11M) spans the UK and is nested within the 12km regional model (HadREM3-GA705), which is in turn driven by the 60km global model (HadGEM3-GC3.05) (Andrews et al., 2019; Williams et al., 2018). The 2.2km-projections are available for three 20-year periods of 1981-2000, 2021-2040 and 2061-2080. Known atmospheric GHG concentrations are prescribed as forcings to the historical 20-year period. For the second and third periods, the projections employed follow the RCP8.5 scenario, which assumes substantial on-going human burning of fossil fuels. The 2.2km projections consist of an ensemble of 12 members (**Table 3**), each of which can be considered as a plausible realisation of the climatic response to rising GHG levels. The local members are driven by different members of the global coupled model ensemble, and corresponding regional model ensemble, created by perturbing uncertain parameters in the model physics within their bounds of uncertainty. Thus, the range of the 2.2km projections provides an estimate of the uncertainty in future changes due to natural variability while additionally accounting for uncertainty in the physics of the driving global climate model. We computed regionally-averaged UKCP temperature and precipitation projections for each of the three regions shown in **Figure 1d**, and for each of the crop growth stages indicated in **Table 1**. For a detailed discussion of modelling assumptions and limitations see section 2.6.

**2.4 Bias correction**

Given the driving parent model of each UKCP Local simulation comes from a perturbed physics ensemble, each ensemble member is typically regarded as a different model and therefore is independently bias-corrected. UKCP Local simulations of area-averaged precipitation and temperature were bias-corrected against the 5km area-averaged observed daily HadUK data (Hollis et al., 2019) for each geographical region, using the entire the historical period of Dec 1980 to Nov 2000 (**Table 3**). The bias correction scaling factors were identified and applied with the "hyfo" (Xu, 2020) package written in the software R. This bias correction approach is a simple scaling method which is additive for temperature and multiplicative for precipitation (one correction factor per ensemble, per region), so it preserves an absolute or relative trend, respectively. The UKCP data have 30 days in each month, therefore, to perform the bias correction we added calendar days for each of the three 20-year

periods (e.g. from 1980-12-01 to 2000-11-30 with only 30 days in each month) and merged the historical period with observed data, removing any non-matched days (e.g. dropping the 31[st] of the month from the observed data, or dropping February 29[th]-30[th] from the projections). This produced two overlapping time series of equal length over the period of Dec 1980 to Nov 2000 to perform the bias correction. We make the assumption these present-day biases are likely to extend into the future periods, a key caveat of any bias correction method. The bias correction factors are relatively small, which suggests the simulations are

well-aligned with the historical observations: x0.89 on average for precipitation for the three regions (individual factors for each member and region are shown in **Table 3**); −0.04°C for minimum daily temperature, +0.54°C for mean daily temperature, and +1.14°C for maximum daily temperature. We apply the bias corrections to the two future UKCP periods (Dec 2020 to Nov 2040 and Dec 2060 to Nov 2080, recalling these are for the RCP8.5 scenario). The bias correction performs well at the annual scale (**Figure 2**) but may differ across specific growth stages and regions (e.g. in the Foundation phase, median

precipitation is slightly overestimated in EMYH and SEE regions) (**Figure 3**). Bias-corrected projections inevitably contain some uncertainty, and should be considered as providing general directions of change.

**2.5 Statistical approach**

We first assess the association between climate metrics and crop yield by using pairwise two-variable Pearson correlations (expressed as annual crop yield versus each individual seasonal climate variable). The magnitudes of the correlation

coefficients and their *p*-values are provided in **Table 2**.

Second, to assess the additive or offsetting effects of different climate conditions across crop growth stages, we develop a multiple linear regression model between regional crop yields and climate (**Equation 1**). We develop one model per region, with different observed temperature and precipitation variables for each region. Using **Table 2**, we purposely select just two continuous variables per growth stage to develop the model (one temperature-based metric and one precipitation-based metric),

thereby avoiding correlated metrics. The equation used to fit the observed data for a given region is formulated as:

$$y = \beta_0 + \beta_1 x_1 + \beta_2 x_2 + \beta_3 x_3 + \beta_4 x_4 + \epsilon \qquad (1)$$

where y represents the wheat yields (t/ha); $x_1$ is Foundation$_{max\_minT}$ (°C), $x_2$ is Foundation$_{total\_P}$ (mm), $x_3$ is Production$_{max\_maxT}$ (°C), and $x_4$ is Production$_{total\_P}$ (mm); $\beta_0$ is the intercept; $\beta_1$ to $\beta_4$ are the regression slope coefficients for each of the explanatory climate variables; and $\epsilon$ is the error. The model statistics and coefficients are provided in **Table 4** for each region.

Although the model is significant (p<0.05) in EMYH, SEE and NAT, the predictability is relatively low. Alternative metrics could also be selected, such as *var_dailyT or var_maxT* in the Production phase, or *days_P>10mm* in either phase, or additional variables reflecting e.g. precipitation intensity. These variables have not been tested and should be evaluated in future research, further developing the statistical crop model. Our model is a proof-of-concept that could be refined to improve the predictive skill if further data becomes available. The Construction phase is not included in the regression model as it shows no consistent

associations with wheat yields (**Table 2**). The multiple regression describes the "extremeness" of climate independently for each crop growth stage, and so may account for compound positive and negative climate impacts on wheat yield across a year.

For instance, detrimental climate conditions may have a cumulative impact on wheat yields if occurring across multiple growth stages, such as heavy precipitation events during the Foundation phase (Foundation$_{total\_P}$), followed by meteorological drought during the Production phase (Production$_{total\_P}$). Conversely, poor conditions in one stage may be mitigated by good conditions or agronomic interventions in another stage (e.g. wet weather leading to increased incidence of fungal disease can be mitigated by subsequent increased use of fungicides), and this would be reflected by the regression model.

Third, to assess future changes in crop yields, we drive the same multiple regression model with the bias-corrected projections of the same variables, computed from the 12 members of the UKCP Local simulations. This approach allows us to fuse together the data-driven regression model with the meteorological simulations for higher greenhouse gas emissions. We use the model results to understand how multivariate climate change could lead to compensating or compounding impacts on future crop yields.

### 2.6 Assumptions and limitations

One of the advantages of the empirical data-driven approach herein is that there are fewer assumptions than in a process-based model approach. However, such an approach makes some key assumptions nonetheless, listed here:

(1) To assess the impact of extreme weather on crop yields, we assume that the crop yields are affected by weather within the pre-defined crop growth stages described in **Table 1**. We employ fixed-in-time growth stages for practicality, but in reality these growth stages may be weather dependent from year-to-year, as plant vulnerabilities to extreme temperatures or precipitation may differ, e.g. from one July to another July. We did not use the 99 detailed physiological growth stages (AHDB, 2022), but rather the high-level growth stages which are defined over long time periods to split the year into key stages of wheat growth.

(2) A major assumption in our regression-based approach is that wheat responses to climatic variables in the past are a reliable predictor of responses in the future. One important uncertainty that we do not consider is how wheat growth and water use might respond to increases in atmospheric $CO_2$ (Ewert et al., 2002; Swann et al., 2016).

(3) Spatially, we average the climate metrics over the three regions. This aggregation to regional scales may mask variation in the weather conditions occurring in individual grid cells (or farms) – for instance the regional average could show little change, but this could hide large local changes (such as less frequent but more intense bursts of rain), or contrasting directions of change within the region. Other spatial metrics, such as extracting the highest rainfall event within each region, may be worth testing in future work.

(4) The multiple regression model describes the impact of compound climate effects in different growth stages on wheat yields but not that of antecedent conditions (memory effects). Compound effects are captured well by our model, e.g. frost conditions during the Foundation phase and heavy waterlogging during the Production phase might combine to produce poor conditions across the whole year. However, the model cannot assess whether the climatic impacts during the Production phase are the

same irrespective of "memory" impacts from the climatic conditions in the earlier plant development stages (for example, the antecedent effects of a warm winter and wet spring in leading to a crop failure, e.g. (Ben-Ari et al., 2018)).

225 (5) For the future projections of altered meteorological conditions, the UKCP18 HadGEM3 climate model simulations (in which the UKCP Local 2.2km simulations are nested) were only performed for the RCP8.5 pathway for atmospheric greenhouse gas concentrations, and we do not address emissions uncertainty from other scenarios. While the likelihood of such high on-going emissions is now considered low (Chen, D. et al., 2021; Hausfather and Peters, 2020), the RCP8.5 scenario is commonly used to facilitate detection of climate signals in future projections above natural variations in the climate (due to 230 the large changes projected), and was deliberately chosen as the configuration for UKCP Local simulations to maximise the signal to noise. Using a high emissions scenario also has the advantage that one can make estimates of climate changes for lower emissions scenarios using scaling approaches.

(6) Our analysis employs one single model, the UKCP Local (2.2km) climate projections. As described in section 2.3, the UKCP Local simulations are driven by a perturbed physics ensemble (PPE) of a single forcing Earth System Model (ESM), 235 i.e. the parameters within the physics of the driving ESM are perturbed within their bounds of uncertainty. Thus, the 12 members of the high-resolution ensemble describe both internal climate variability and the climate modelling uncertainty in the driving model (i.e. they have wider uncertainty than is typically represented in one single climate model). The trends of the UKCP Local simulations therefore at least partially cover the range of uncertainty and trends that would occur in the ESMs developed by other climate research centres. However, the climate modelling range of uncertainty is likely to be underestimated 240 since the UKCP Local ensemble lacks information from other international climate models. In winter, the UKCP Local simulations show some higher precipitation responses compared to the full CMIP5 ensemble due to the improved representation of winter-time convective showers in the Local 2.2km model (Kendon et al 2020). UKCP Local projections also project relatively high temperature changes compared with other climate models (see e.g. https://interactive-atlas.ipcc.ch/regional-information). Changes in summer precipitation show a considerable drying in the UKCP Local 245 projections, whereas the CMIP5 and CMIP6 simulations indicate that outcomes with more modest reductions or small increases should also be considered.

(7) The UKCP Local projections provide high spatial resolution (2.2km) downscaling of global climate model projections specifically for the UK. Such high resolution simulations are able to at least partially resolve convective storms, and do not require a parameterisation scheme to provide a representation of convection, which is a simplification of the real world and a 250 known source of model deficiencies. These simulations are therefore considered more reliable for projecting future changes in rainfall characteristics. However, there is still uncertainty in the parameterisation of UKCP Local, and so it can be expected that as future research groups also build convective-permitting models, differences will emerge that we are presently unable to account for.

## 3 Results and Discussion

### 3.1 Historical increases in wheat yields and interannual yield volatility

Since the late 1800s, and especially since the 1950s, there has been exceptional growth in UK wheat yields due to rapid advances in crop breeding, increasing farm mechanisation and the availability of agrochemical inputs, such as fertilisers (**Figure 1a**). Sustained increases throughout the 1980s-90s reflect the development of farming technologies, varieties, improved nutrient use efficiency and effective pesticides and growth regulators. Available time series of crop yields are much shorter when disaggregated to the regional level (**Figure 1b**) than at the national level (**Figure 1a**). Of particular note, though, is that the EMYH and SNE regions exhibit some levelling of wheat yields since 1990, mirroring the national trend, while the southernmost region, SEE, has seen some continued increases (**Figure 1b**).

In addition to increases in mean yields, the national yield time series exhibits a visible increase in the variance of yields in the last few decades (**Figure 1c**). This increase in volatility is not solely driven by increases in the mean of the time series. A comparison of the variance of crop yields between the periods 1885-1989 (105 years) and 1990-2020 (31 years) using both Levene's test (p=0.022) and the non-parametric Fligner-Killeen's test (p=0.093) indicates that there is a significant difference in the variance. The results are even more significant when comparing periods of similar length, 1960-1989 and 1990-2020 (30-31 years) for both Levene (p=0.002) and Fligner-Killeen (p=0.003), or focussing on the last two decades, 1970-1999 and 2000-2020 (30-21 years); p<0.001 for both tests. A question of notable interest, therefore, is understanding why the variance of yields has significantly increased, and whether it might be associated with more frequent or intense weather extremes.

### 3.2 Association between climate extremes and wheat yields in each crop growth stage

We assess the association between seasonal climate and crop yields by using precipitation and temperature metrics during the three crop growth stages. We expect the association between climate anomalies and wheat yields to differ regionally due to a range of factors, including the resilience of the wheat plant, husbandry practices of farmers and agronomists, biophysical conditions (e.g. soils, day length), and climatic differences (e.g. rainfall tends to be more frontal in the north, with orographic rainfall over high ground, and more convective in the southeastern UK). Although only some of the associations between the seasonal climate metrics and annual crop yields are statistically significant, we show all the associations and their relative strength for full transparency (**Table 2**). In **Figures 4-5**, we focus on *total_P*, *max_minT*, and *max_maxT* in each growth stage, as these are some of the most consistent metrics in the historical data (**Table 2**); figures produced using *max_maxT* give very similar patterns to *max_meanT* (not shown). These figures reveal the climatic 'space' generated by the interaction between precipitation and temperature in each growth stage: some of the worst UK wheat yields in recent decades have occurred during years with anomalously high/low seasonal rainfall, or prolonged heat, an important indicator of crop heat stress (Arnell and Freeman, 2021). The figures also indicate that temperature and precipitation are not independent from one another, since the wet years with poor yields also tend to be relatively cool (e.g. 2001, 2020 in the Foundation phase), and the dry years can sometimes be particularly hot (e.g. 1976, 2018 in the Production phase) (**Figure 5**).

From a crop physiology perspective, in the Foundation phase (October to early April; **Table 1**), prolonged waterlogging of the soil may suppress wheat yields by restricting root development and plant growth (AHDB, 2018). We find a significant negative association between crop yields and the number of heavy rainfall days in the EMYH region (**Table 2**, *days_P>10mm*); as can be seen in Figure 5 (years 2001, 2020; **Figure 4a**). The association between yields and total_P *days_P>10mm* and yields is also negative in SEE and at the national scale, but not significantly so. In the winter of 2000/01, for instance, wet autumn and winter conditions resulted in delayed sowing and poor seedbed conditions. Additionally, colder-than-usual conditions in the Foundation stage (e.g. year 2013, not shown) may delay or prevent crop tillering: frost can damage early drilled and fast-growing varieties, while frost heave can kill seedlings. We find significant positive associations between yield and *max_minT* at the national scale and in the EMYH region, and with *min_meanT* and *min_minT* in the SEE region (**Table 2**). The positive associations indicate that warming temperatures may benefit UK wheat yields in a warming climate.

While crops are growing rapidly during the Construction phase (April to early June), both late frosts and dry weather can reduce crop growth (**Table 1**). For this period in each year, we find no significant associations between climate characteristics and crop yields (**Table 2**). This is not necessarily a contradiction, as reduced growth does not always carry through to reduced yield. Both low yields (e.g. years 1976, 2001, 2020; **Figure 4b**) and some high yields (1962, 1984) have occurred during drier-than-average Construction phases. Overall, wheat yields seem to be more sensitive to climate conditions during the Foundation or Production phases.

The clearest association between climate extremes and crop yields seems to be in the Production phase, which is the time from post-flowering to harvest (summer: June and July). It is during this phase that yields may be susceptible to both drought and water logging (**Table 1**). We find a consistently negative association between heavy rainfall (both *total_P* and *days_P>10mm*) and crop yield in all three regions. For *total_P* the association is significant in EMYH and at the national scale, and for *days_P>10mm* in EMYH (**Table 2**). The association between low wheat yields and high summer rainfall is apparent in specific years such as 2007 and 2012 (top left of **Figure 4c** and **Figure 5c**). For example, year 2012 witnessed exceptionally poor yields due to high spring and summer rain, a high incidence of fungal disease (e.g. *Septoria tritici*) (DEFRA, 2012) and low sunlight during the grain-filling period (i.e. the first part of the production period, when the grain is swelling and requires sunlight for photosynthesis). In contrast, good yield years are often associated with warm summer temperatures and moderate to low rainfall: this can be seen in the positive associations between wheat yields and *max_maxT* or *max_meanT,* which are significant both nationally and in EMYH (**Table 2**). Examples are years 2015, 2019 (**Figure 4c, Figure 5c**). During the Production phase, meteorological drought conditions may also have negative impacts. Hot, dry weather shortens the growth period, resulting in early canopy senescence and reduced grain weight (**Table 1**). Indeed, some of the UK's poorest crop yields occurred during warm, dry summers (e.g. years 2013, 2018 in SNE and SEE; **Figure 5c**). The benchmark grain-filling period is 45 days from flowering until maximum dry weight in late July, but it can be as short as 28 days during severe droughts (AHDB, 2018).

### 3.3 Explaining the association between crop yields and climate extremes: compound impacts across growth stages

It can be challenging to systematically identify the weather conditions to which wheat yields are most vulnerable within individual growth stages. Most of the correlations in the historical data are not statistically significant (**Table 2**). The often relatively weak association between climate anomalies and wheat yields at the level of individual growth stages can be explained partly by the shortness of observational records, the combined resilience of the wheat plant (i.e. physiological reproductive mechanisms) and the husbandry skills of farmers and agronomists in mitigating these impacts by adjusting to climatic extremes. There is thus a role for agronomic management in mitigating apparent relationships with climate: this role might not be as direct as irrigating in response to drought, but farmers can dampen the effects of climatic variation through crop management, for example, by changing fungicide regimes to response to increased fungal disease brought about by wetter conditions, changing the timing or amount of inputs of nutrients, pesticides and growth regulators (Knight et al., 2012). The relatively intensive nature of UK wheat production (Hillocks, 2012; Wesseler et al., 2015) may thus be sufficient to dampen crop responses to climatic variation (Gagic et al., 2017). Farmers can also change many other aspects of management, including wheat variety, tillage, sowing date, sowing rate, or harvest date, in response to forecast or current conditions. Wheat cultivars are bred with a measure of resistance to certain climatic variables, so a farmer can select a cultivar appropriate to local climatic conditions (Kahiluoto et al., 2019).

Low correlations between climate and yield anomalies over seasonal wheat growth stages may also reflect compensatory effects between growing phases. For instance, a less than ideal Foundation phase might be offset by a favourable Production phase or *vice versa*. It is equally important to note that growing phases in real plants are determined by their growth, rather than calendar days. Thus a phase may last longer, resulting in delayed crop growth, but maintaining the expected yield. Our calendar-fixed phases are a simplification of this process.

Conversely, cumulative detrimental impacts of climate across stages (e.g. accumulated rainfall and subsequent waterlogging) may be one of the most damaging factors affecting overall annual crop yields. In other words, the flexibility and techniques farmers have at their disposal to adapt to climate variability are bounded. For instance, low yields in year 2018 were due to very dry conditions in the Foundation stage, followed by very hot and dry conditions in the Construction and Production stage (DEFRA, 2018a). In contrast, very low yields in years 2001 and 2007 were caused by a combination of high rainfall in the Foundation and Production stages (**Figure 4**). The exceptionally wet winter of 2019 (affecting the 2020 harvest in **Figure 4**) also imposed severe constraints on farming operations and resulted in a reduction in the areas of autumn-sown crops. These examples illustrate why a full understanding of projected changes to temperature and precipitation across wheat growth stages is required.

To try to assess the offsetting or additive effects across growth stages, we develop a simple multiple regression model relating the observed wheat yields in each region to just two metrics reflecting temperature and precipitation conditions in the most important stages based on the outcomes of **Table 2**: Foundation$_{max\_minT}$, Foundation$_{total\_P}$, Production$_{max\_maxT}$, and

Production$_{total\_P}$. We find this model is significant at the 95% level (p<0.05) for EMYH, SEE, and the national scale but not SNE (**Figure 6, Table 4**). The lack of significance in SNE can be easily explained, since the association between yields and Foundation$_{max\_minT}$ is weak there (R=0.15), but good elsewhere (~R=0.3 in SEE and EMYH). Similarly, the association between Foundation$_{total\_P}$ and annual yields is negative in other regions (~R=-0.2/-0.28 in SEE and EMYH) but weakly positive in SNE (**Table 2**). The multiple regression model displays the best fit in the EMYH region, where the climate metrics display

the strongest correlations with yields (significant for Foundation$_{max\_minT}$, Production$_{max\_maxT}$, and Production$_{total\_P}$). As expected, these model fits only explain a portion of yield variation (R2 ranges from 0.12 for SNE to 0.32 for EMYH), since crop yields are only partially explained by climate, as discussed above. However, the models allow us to capture the multivariate impacts of temperature (Foundation$_{max\_minT}$ and Production$_{max\_maxT}$ exhibit a positive association with crop yields) and precipitation (Foundation$_{total\_P}$ and Production$_{total\_P}$ exhibit a negative association with yields). Thus, the strongest associations between

climate and yield anomalies may occur during years with cumulative climate impacts across growth stages. Cumulative impacts can be seen in the improved R2 in the multiple regression compared to the pairwise correlations. In other words, the model is capturing something individual variable correlations do not, and this could be that compensation across phases. That said, whether this added explanatory power is from inter-stage compensation, or compensation between variables within a single stage, is not clear from the regression results alone.

**3.4 Annual projections of future climate conditions and implications for crop yields**

    At the annual scale, projections of future maximum hourly temperature are available for the periods 2021-2040 and 2061-2080 from the UKCP Local simulations. The interquartile range of projected temperature for 2021-2040 lies well above the median of historical extremes (**Figure 2a-c**). Future high-temperature conditions generally fall beyond the bounds of annual variability experienced in the contemporary period for all three wheat-growing regions (**Figure 2c**). As expected, changes are largest for

the later modelled period 2061-2080, corresponding to higher atmospheric greenhouse gas concentrations. This exceedance of historical thresholds by temperature projections is true for all 12 UKCP Local ensemble members, independent of uncertainty in changes in the large-scale conditions sampled by perturbing parameters in the Hadley Centre global climate model. However, it is important to note that the 12 climate model members (**Table 3**) do not sample the full range of uncertainty, evident in differences between all available global climate models (Kendon et al., 2021); see section 2.6 for a discussion.

For total annual precipitation (**Figure 2d**), the projections do not indicate a very obvious increase or decrease in any of the three regions relative to the historical period, although SNE may seem very slightly wetter, and SEE very slightly drier on average (comparing medians) in the later period (2061-2080). This lack of trend in yearly data may be explained by the opposing changes in the different seasons: in general the winter season is projected to become wetter and the summer drier (Kendon et al., 2021). Importantly, there are also changes in the underlying intensity and frequency of precipitation (e.g.

significant increases in *days_P>10mm* and *var_P* in the Foundation phase, **Figure 7**), which are not evident from simply looking at trends in annual mean precipitation.

### 3.5 Seasonal projections of future wheat-growing conditions and crop yields

When considering UKCP Local projections by wheat growth stages (instead of at the annual scale), clearer patterns become apparent (**Figure 3**). We expect to find spatial differences in the climate projections, as they exhibit north-south gradients in changes across the UK. Even in a single ensemble, there are gradients in the future changes in rainfall which differ from present-day climatology and relate to regional differences in increases in moisture availability as well as changes in circulation patterns. The question of compound climate change – i.e. the joint impacts of temperature and rainfall, or moisture availability – is important for future crop yields.

Contrary to global expectations of declining yields under climate change, the multiple regression model indicates that projections of future temperature and precipitation change are likely to contribute to a continued growth of future wheat yields in the UK (**Figure 6**). These projections rely on broad estimates of changing night/day temperature extremes as well as total rainfall in the Foundation and Production stages. It is possible that more data may provide greater information on changing water availability, atmospheric vapour demand, and plant stress; however with the existing observations our data-driven approach highlights that a changing climate may not be entirely negative for wheat yields. This can be explained as follows.

For the Foundation phase (October to early April), all regions can expect to see progressively warmer, wetter conditions in the coming decades according to the UKCP simulations. Significant projected increases in *max_minT*, *max_maxT*, and *total_P* are evident in all three regions (**Figure 7**). Such conditions might not necessarily adversely affect wheat production (**Figure 4**), and are likely to be beneficial in decreasing the risk of frost damage (**Table 2**). When considering *max_minT* and *total_P*, the projections indicate that there is a good chance of seeing more temperate winters similar to the one preceding year 2015, where Foundation conditions were warm and not too wet, resulting in high crop yields (**Figure 4a**); however, the significant projected increases in total rain, heavy rain, and rainfall variability (*total_P*, *days_P>10mm* and *var_P*; **Figure 7**) in all three regions may equally prove problematic beyond certain thresholds. In very wet years, the UK may also experience winters more like those of 2001 and 2020, which led to low yields across the UK (**Figure 4a**), especially in EMYH/SEE (**Figure 5a**).

Projections for the Construction phase (mid-April to mid-June) are not included in the multiple regression model, due to the lack of significant associations between climate and wheat yields (**Table 2**). During this phase, the projections indicate significant decreases in *total_P* in EMYH and SEE, but not SNE (**Figure 7**). There are no evident changes in heavy rain (*days_P>10mm*; **Figure 7**), and we find considerable overlap with both good and poor yields in the historical data (**Figure 4b**). These findings suggest that the Construction phase may not necessarily be the most at-risk in terms of the impacts of changing UK climate to crop yields. Although precipitation may not change much, there is still warming, which will increase atmospheric vapour demand (all else being equal). Thus, understanding the effects of compound change such as heat waves and drought (Zampieri et al., 2017) or the evaporative role of temperature (Lobell et al., 2013) is important to help provide more robust conclusions about the future.

In the Production phase (mid-June to end of July), UKCP simulations project both much warmer and drier conditions (**Figure 3, Figure 7**). The drying signal is relatively similar across the three regions and becomes more apparent in the later simulations towards the end of the century. It is important to note that the UKCP Local projects stronger drying than CMIP5-6 models. Projected trends also indicate significant, strong increases in *max_minT*, *max_maxT*, and equally in temperature variability (*var_dailyT* and *var_maxT*; **Figure 7**). A simple analogue approach suggests we may see more Production phases similar to years 2006, 2015 and 2019 in the EMYH/SEE regions, conducive to high yields (**Figure c**). Both the national and the regional data suggest all regions may benefit from a warmer and drier Production phase (**Figure 4c-5c**). The projected trends reveal significant decreases in rainfall total and variability (*total_P* and *var_P*) in all three regions but no apparent decreases in heavy rain (**Figure 7**). However, individual anomalous years with poor yields and warm dry conditions remain plausible, such as year 1976 at the national scale (**Figure 4c**), and 2013 in the SNE and SEE areas (**Figure 5c**). Because the projected high-temperature conditions are outside those experienced in the historic period, there is also a risk that the positive association between hotter, drier Production phases and enhanced yield found in the historical observations will no longer hold. This is especially true since temperature could have non-linear impacts (e.g. sterility or abortion of formed grains) through the physiological effects of frost and heat shock (Barlow et al., 2015). Droughts and heatwaves severe enough to have a substantial impact on yield are rare in the historic data (Knight et al 2012), and so we have little data by which to determine at what thresholds temperature and dryness cease to be beneficial for wheat and begin to have negative impacts. However, the anomalous years (e.g. 1976 and 2013) suggest that this can occur, and recent research indicates that days exceeding heat stress temperatures for wheat are likely to increase under climate change (Arnell et al., 2021).

Overall, projections of future temperature and precipitation conditions suggest a continued increase of future wheat yields when relying on *max_minT, max_maxT* and *total_P* (**Figure 6**). The higher yields are found in the far-future period (2061-2080) partly due to the effect of warming conditions and thus reduced frost risk in the Foundation phase. These beneficial impacts may however be offset by the significant increases in heavy rainfall (and rainfall variability) projected in the Foundation phase and enhanced meteorological drought conditions in the Production phase (**Figure 7**). The offsetting between climate effects, e.g. the interactions between temperature and precipitation, is an important mechanism and uncertainty both in the climate and in terms of their implications for crops. For instance, very hot conditions in the UK can often only be reached with a dry land surface (visible as apparent negative temperature-precipitation correlations during the Production stage, **Figures 4-5**). Drought and heatwaves are believed to self-intensify and propagate due to feedbacks between the land and atmosphere (Miralles et al., 2019). Cool and wet conditions could also be linked physically (e.g. Production phase in 2007 and 2012), with implications for crop yields. This raises questions about joint heat and moisture impacts and how their interdependence might evolve into the future as greenhouse gases rise.

Lastly, the impact of rising atmospheric $CO_2$ on crop water use is an important uncertainty which is hard to model via our statistical approach and likely to impact future crop growth (Ewert et al., 2002; Swann et al., 2016), but is not considered herein. Overall, our approach suggests that, on average, climate change is likely to have more positive impacts on UK wheat

yields than previously considered. However, against this background of average positive change, our work illustrates that we are likely to move outside of the climatic envelope which wheat farming in the UK has previously adapted to. Thus, the new weather conditions generated by the effects of rising temperatures (including intense local thunderstorms) are only likely to increase the degree to which farmers may struggle to mitigate against climate impacts.

## 4 Conclusions

Mean UK crop yields saw a rapid growth in the 1950s followed by a plateau in the 1990s, then substantial increases in the inter-annual variability of yields. This acceleration has been challenging for UK wheat farmers, since crop yields over the past two decades (2000-2020) have been significantly more volatile than over the previous century (**Figure 1**).

A first question is thus our ability to explain such changes, and assess whether statistically significant associations exist between observed temperature/precipitation metrics and historical wheat yields during the three crop growth stages, in the three main wheat-growing regions of the UK. While the plateau in yields can be explained by a variety of technological and agronomic factors (Knight et al., 2012), we find some evidence that yields over the last 30 years can be partially explained by climate metrics such as warm night temperatures and heavy rainfall days in the Foundation phase (principally in the EMYH region), or maximum daily temperatures, daily temperature variability and total precipitation in the Production phase (**Table 2**; with correlation strength and significance varying regionally). Significant statistical associations are found principally in the Foundation and Production phases and for regions EMYH and NAT. Yields are more fully explained when considering a multiple regression model (**Figure 6**) characterising additive and offsetting impacts of climate across growth phases (e.g. detrimental impact of very cold temperatures in Foundation phase followed by very high precipitation in the Production phase). However, it is unclear whether the added explanatory power of the regression model is from inter-stage compensation, or compensation between variables within a single growth stage. This would be an area for further research. The data-driven regression could additionally be refined by including various thresholds (e.g. considering the beneficial impacts of a warm and dry Production phase only up to certain limits relevant to plant stress). We find the association between historical climate and crop yields is most evident in years which saw compound extremes (Zscheischler et al., 2020), i.e. climate anomalies across multiple growth stages (e.g. 2007, 2012, 2020, **Figures 4-5**), 'escaping' the ability of farmers to adapt through agronomic means. Outside these combined extremes, the data indicate a strong inter-annual resilience of wheat production, implying that at present farmers can, and do, successfully utilise crop husbandry to maintain yield levels.

Our second question seeks to understand how projections of compound temperature and precipitation extremes might impact future crop yields under a high-emissions climate scenario. Overall, the data provides a surprisingly favourable outlook of climate conditions for future crop yields. During the Foundation phase, high seasonal values of night temperatures ($max\_minT$) are correlated positively and significantly with crop yields in EMYH and nationally (**Table 2**), suggesting that the significant future increases projected by the UKCP Local simulations (**Figure 7**) are likely to provide more beneficial growing conditions

during the winter. These positive temperature impacts may be offset by the significant projected increases in rainfall total, heavy rainfall, and rainfall variability in all three regions (*total_P*, *days_P>10mm*, and *var_P*, **Figure 7**), since heavy rainfall is detrimental to wheat yields (in EMYH especially; **Table 2**). Later in the year during the Production phase, when high day

temperatures are significantly and positively associated with wheat yields in EMYH and nationally (**Table 2**), the UKCP local simulations also project significantly warmer and drier mean conditions (**Figure 7**), which may be conducive to positive yields, similar to the years 2015 and 2019 (**Figure 4**). Since high rainfall totals in the Production phase adversely affect growing and production conditions (*total_P* is negatively and significantly associated with crop yields in EMYH and nationally, **Table 2),** the projected significant decreases in future rainfall (which are stronger in UKCP Local than in CMIP5 and 6) could equally

be beneficial to wheat yields (*total_P*, **Figure 7**). Future anomalous years similar to 2020, with a wet crop Foundation phase and a much drier Construction phase that significantly suppressed yields (**Figure 4**), are a possibility (**Figure 7**). It seems plausible that the farming community may also face increased inter-annual variability in the future, e.g. a sequence of dry years (similar to 2019) followed by very wet years (2001, 2012) against a backdrop of warmer and wetter/drier conditions. Further analyses could equally assess whether the optimal time and place to grow wheat is changing, or the effects of changes in

rainfall patterns at the local (rather than regional) scale.

In summary, this work provides evidence that wheat yields over the last 30 years are associated with combined temperature and precipitation extremes, especially across the crop Foundation and Production phases, in the EMYH region and nationally (**Table 2**). Although the climate projections provide a generally positive outlook for future yields across the UK, it is important to note that the relationships between past wheat yields and historic climatic conditions may not be adequate guides to the risks

associated with projected future conditions, as future temperature extremes and rainfall lie outside the range of conditions that UK agriculture has so far experienced. Further, this work studies climate extremes at the regional scale, but not local changes in rainfall intensity and variability, which are beyond the scope of the paper (e.g. drier average regional conditions may hide less frequent but more intense local thunderstorms). Out of caution, therefore, a priority is to continue developing resilient agricultural systems to emerging climate patterns, as the global demand for wheat and other crops has been projected to double

from 2005 to 2050 (Tilman et al., 2011). As higher-resolution crop yield data become available, further research into robust process-based or AI-informed crop models, alongside improved collaboration across spatial, governance and supply-chain scales (Holman et al., 2021), will be required to help farmers adapt to evolving climate conditions and maintain the security of wheat production.

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

## Acknowledgements

We gratefully thank Corey Lesk and two other anonymous reviewers for their helpful comments, which greatly improved the manuscript. L.J.S gratefully acknowledges funding from the UK Research and Innovation FLF scheme (MR/V022008/1). C.H., R.F.P. and J.R. gratefully acknowledge the Agland project. C.H. also acknowledges the NERC CEH National Capability Fund. R.F.P., C.H. and J.R. were supported by research programme NE/N018125/1 LTS-M ASSIST - Achieving Sustainable Agricultural Systems, funded by NERC and BBSRC. E.J.K. gratefully acknowledges funding from the Joint UK BEIS/Defra Met Office Hadley Centre Climate Programme grant no. GA01101.

## Code/Data Availability

All data employed in the manuscript are publicly available as described in the Methods. (1) HadUK gridded 5km observational temperature and precipitation data were obtained from the National Climate Information Centre. (2) UKCP 2.2km Local temperature and precipitation projections were obtained from the UK Met Office. (3) UK wheat yields were obtained from the UK Department for Environment, Food and Rural Affairs (DEFRA). The code to produce the analyses can be obtained from the corresponding author upon reasonable request.

## Author contributions

L.J.S. led the coding, analysis and visualization. C.H. extracted the regional UKCP Local projections. All authors contributed to designing the experiments and writing the manuscript.

## Competing interests

The authors declare that they have no conflict of interest.

**Table 1: Three standardised wheat growth stages, modified by one day to avoid overlap across stages (AHDB, 2018).**

| Growth Stage | Benchmark start date | Benchmark end date | Potential climate impacts on the crop |
|---|---|---|---|
| Foundation phase | 1st October | 9th April | Crop is germinating and growing slowly. Susceptible to waterlogging and frost damage |
| Construction phase | 10th April | 10th June | Crop is green and growing rapidly. Needs adequate light, can be affected by late frosts |
| Production phase | 11th June | 26th July | Period of post-flowering to harvest, grains fill and ripen. Susceptible to drought and waterlogging |

**Table 2: Association between observed climate metrics and wheat yields in each crop growth stage and region. Table indicates Pearson's correlation coefficients and their p-values (\*\*\* indicates p<0.01, \*\* p<0.05, \* p<0.10). National data is tailored to the same time period as regional data here (31 years between 1990-2020) for comparability. Note: *total_P* and *mean_dailyP* are equivalent. Some of the most relevant metrics with relatively consistent sign are indicated in bold font (see Figure 7 for trends in these metrics).**

| | | | Foundation | | | | Construction | | | | Production | | | |
|---|---|---|---|---|---|---|---|---|---|---|---|---|---|---|
| | | | SEE | EMYH | SNE | National | SEE | EMYH | SNE | National | SEE | EMYH | SNE | National |
| Maximum daily temperatures | Quantiles of the region-averaged maximum daily temperature across the phase/year (e.g. *max_maxT* is the highest daily maximum temperature) | *max_maxT* | 0.01 | 0.01 | 0.10 | 0.03 | 0.11 | -0.27 | -0.11 | -0.19 | **0.26** | **0.42\*\*** | **0.22** | **0.42\*\*** |
| | | *mean_maxT* | 0.08 | 0.05 | 0.16 | 0.08 | -0.06 | -0.09 | 0.09 | -0.10 | 0.14 | 0.22 | 0.24 | 0.20 |
| | | *min_maxT* | 0.17 | 0.01 | 0.01 | 0.04 | 0.05 | 0.09 | 0.14 | 0.00 | 0.06 | -0.06 | 0.09 | -0.01 |
| Mean daily temperatures | Quantiles of the region-averaged mean daily temperature across the phase/year (e.g. *max_meanT* is the highest daily mean temperature) | *max_meanT* | 0.24 | 0.21 | 0.12 | 0.24 | 0.23 | -0.16 | -0.26 | -0.11 | 0.23 | 0.41\*\* | 0.14 | 0.46\*\*\* |
| | | *mean_meanT* | 0.06 | 0.05 | 0.16 | 0.09 | -0.03 | -0.07 | 0.07 | -0.07 | 0.07 | 0.11 | 0.16 | 0.11 |
| | | *min_meanT.* | 0.30\* | 0.02 | -0.03 | 0.09 | -0.08 | -0.21 | 0.07 | -0.23 | 0.17 | -0.03 | 0.07 | -0.01 |
| Minimum daily temperatures | Quantiles of the region-averaged minimum daily temperature across the phase/year (e.g. *max_minT* is the highest minimum temperature) | *max_minT* | **0.29** | **0.30\*** | **0.15** | **0.35\*** | -0.06 | -0.19 | -0.16 | -0.14 | 0.18 | 0.28 | 0.05 | 0.27 |
| | | *mean_minT* | 0.03 | 0.05 | 0.16 | 0.09 | 0.02 | -0.03 | 0.05 | -0.03 | -0.07 | -0.11 | -0.01 | -0.08 |
| | | *min_minT* | 0.31\* | 0.11 | -0.01 | 0.07 | 0.00 | -0.26 | -0.02 | -0.24 | 0.18 | 0.00 | 0.02 | 0.12 |
| Daily temperature variability | Mean daily temperature variability (daily maximum - minimum) over the phase/year | *var_dailyT* | 0.13 | 0.03 | 0.06 | 0.02 | -0.10 | -0.11 | 0.07 | -0.11 | **0.22** | **0.36\*\*** | **0.34\*** | **0.32\*** |
| Seasonal temperature variability | Intra-phase/annual variability (max-min) of the max, mean or minimum daily temperatures (e.g. difference between the highest/lowest maximum daily temperature) | *var_maxT* | -0.10 | 0.00 | 0.07 | 0.00 | 0.05 | -0.29 | -0.20 | -0.14 | **0.21** | **0.42\*\*** | **0.16** | **0.41\*\*** |
| | | *var_meanT* | -0.06 | 0.13 | 0.09 | 0.09 | 0.22 | 0.06 | -0.25 | 0.11 | 0.09 | 0.36\*\* | 0.08 | 0.41\*\* |
| | | *var_minT* | -0.05 | 0.09 | 0.09 | 0.17 | -0.04 | 0.06 | -0.10 | 0.08 | -0.05 | 0.25 | 0.02 | 0.10 |
| Precipitation magnitude | Total region-averaged precipitation (P) over the phase/year | *total_P* | **-0.20** | **-0.28** | **0.12** | **-0.14** | 0.06 | 0.08 | 0.03 | 0.08 | **-0.27** | **-0.45\*\*** | **-0.27** | **-0.39\*\*** |
| | Quantiles of daily precipitation computed across the phase/year (e.g. *max_dailyP* is the highest daily precipitation) | *max_dailyP.* | 0.08 | -0.44\*\* | 0.17 | -0.18 | 0.07 | 0.07 | 0.07 | 0.16 | 0.02 | -0.34\* | -0.19 | -0.16 |
| | | *mean_dailyP.* | -0.20 | -0.28 | 0.12 | -0.14 | 0.06 | 0.08 | 0.03 | 0.08 | -0.27 | -0.45\*\* | -0.27 | -0.39\*\* |
| Seasonal precipitation variability | Intra-phase/annual variability of daily precipitation | *varP_Q0.95-Q0.05* | -0.15 | -0.32\* | 0.04 | -0.17 | 0.11 | 0.12 | 0.02 | 0.19 | -0.19 | -0.36\*\* | -0.2 | -0.15 |
| Precipitation frequency | Number of days in the phase/year where P exceeds 10 mm (less than 0.01 mm) | *days_P >10 mm* | **-0.23** | **-0.41\*\*** | **0.13** | **-0.18** | 0.00 | 0.05 | -0.01 | -0.02 | **-0.30** | **-0.31\*** | **-0.25** | **-0.16** |
| | | *days_P <0.01 mm* | -0.01 | -0.10 | -0.07 | -0.27 | -0.27 | -0.20 | -0.06 | -0.27 | 0.14 | 0.23 | 0.19 | 0.13 |

Table 3: Bias correction factors for region-averaged total daily precipitation and minimum/mean/maximum daily temperature for each of the three regions (columns) and each of the 12 UKCP ensemble members (rows) relative to HadUK observed data. These are the complete data (ensembles 02, 03, and 14 do not exist in the UKCP Local dataset). Bias correction is performed using daily data over the common historical period 1980-01-12 to 2000-30-11. The bias correction factors are multiplicative for precipitation and additive for temperature.

| | Precipitation | | | Minimum temperature | | | Mean temperature | | | Maximum temperature | | |
|---|---|---|---|---|---|---|---|---|---|---|---|---|
| ensemble | EMYH | SEE | SNE | EMYH | SEE | SNE | EMYH | SEE | SNE | EMYH | SEE | SNE |
| 01 | 0.82 | 0.88 | 0.88 | -0.30 | -0.54 | 0.17 | 0.35 | 0.18 | 0.73 | 0.97 | 0.89 | 1.38 |
| 04 | 0.79 | 0.80 | 0.91 | 0.12 | -0.12 | 0.54 | 0.82 | 0.69 | 1.11 | 1.47 | 1.47 | 1.75 |
| 05 | 0.84 | 0.91 | 0.9 | -0.14 | -0.28 | 0.17 | 0.57 | 0.51 | 0.76 | 1.24 | 1.27 | 1.43 |
| 06 | 0.87 | 0.96 | 0.92 | -0.03 | -0.22 | 0.35 | 0.57 | 0.42 | 0.88 | 1.18 | 1.06 | 1.53 |
| 07 | 0.88 | 0.95 | 0.94 | 0.13 | -0.10 | 0.62 | 0.69 | 0.51 | 1.08 | 1.25 | 1.14 | 1.65 |
| 08 | 0.81 | 0.85 | 0.86 | -0.51 | -0.70 | -0.17 | 0.08 | -0.05 | 0.37 | 0.64 | 0.59 | 0.97 |
| 09 | 0.97 | 1.06 | 0.97 | 0.26 | 0.10 | 0.68 | 0.69 | 0.55 | 1.05 | 1.13 | 1.02 | 1.55 |
| 10 | 0.87 | 0.95 | 0.92 | 0.03 | -0.18 | 0.39 | 0.47 | 0.29 | 0.78 | 0.91 | 0.75 | 1.28 |
| 11 | 0.80 | 0.84 | 0.89 | -0.13 | -0.36 | 0.26 | 0.58 | 0.42 | 0.86 | 1.22 | 1.15 | 1.52 |
| 12 | 0.89 | 1.00 | 0.93 | 1.08 | 0.75 | 1.73 | 1.66 | 1.42 | 2.21 | 2.27 | 2.10 | 2.84 |
| 13 | 0.85 | 0.94 | 0.87 | -0.67 | -0.87 | -0.27 | -0.13 | -0.31 | 0.24 | 0.39 | 0.26 | 0.82 |
| 15 | 0.82 | 0.89 | 0.85 | -1.14 | -1.36 | -0.7 | -0.6 | -0.79 | -0.16 | -0.13 | -0.27 | 0.39 |
| mean | 0.85 | 0.92 | 0.90 | -0.11 | -0.32 | 0.31 | 0.48 | 0.32 | 0.83 | 1.05 | 0.95 | 1.43 |

Table 4: Statistics of the multiple linear regression model (Equation 1) for each region and nationally (historical observed data, 1990-2020). The low R2 values can be explained by the fact that climate is only one of the mechanisms driving crop yields alongside agronomic management, as discussed in section 3.3. Significance of the coefficients is indicated with stars (*** indicates p<0.01, ** p<0.05, * p<0.10). We use these model fits to drive climate-based projections of future crop yields using the UKCP Local ensemble simulations (Figure 6), assuming no future changes in agronomic management practices. The predictions issued by the regional models are similar to those issued by the national model (Figure 6).

| | n years | p-value | R2 | Adjusted R2 | $\beta_0$ (Intercept) t/ha | $\beta_1$ (Foundation max_minT) t/ha/°C | $\beta_2$ (Foundation total_P) t/ha/mm | $\beta_3$ (Production max_maxT) t/ha/°C | $\beta_4$ (Production total_P) t/ha/mm |
|---|---|---|---|---|---|---|---|---|---|
| EMYH | 31 | **0.032** | 0.324 | 0.220 | 5.9990*** | 0.1066 | -0.0013 | 0.0513 | -0.0031 |
| SEE | 31 | **0.043** | 0.306 | 0.199 | 3.8128** | 0.1480* | -0.0001 | 0.0753 | -0.0014 |
| SNE | 31 | 0.506 | 0.116 | -0.020 | 5.3988*** | 0.0703 | 0.0006 | 0.0661 | -0.0013 |
| NAT | 31 | **0.010** | 0.390 | 0.296 | 3.9199** | 0.1404** | -0.0001 | 0.0921* | -0.0015 |

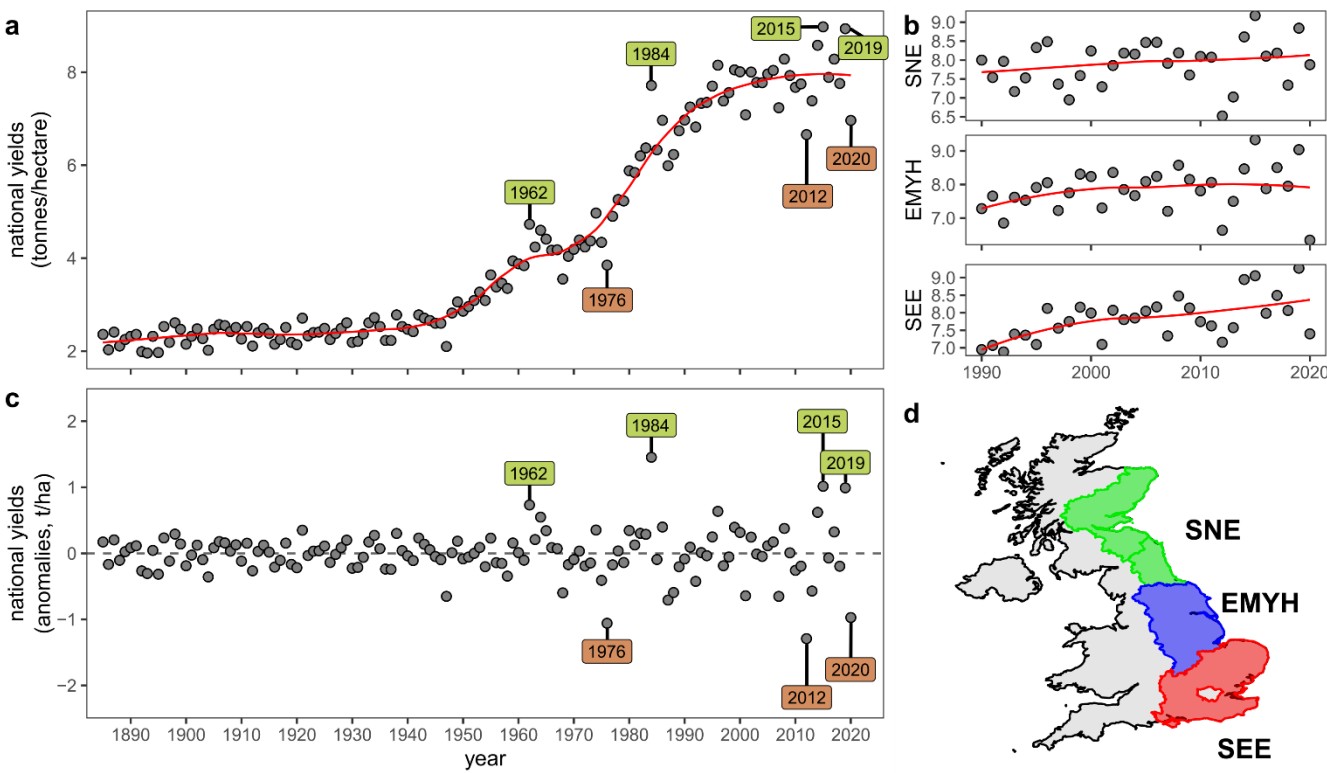

**Figure 1: UK national and regional wheat yields. (a)** National wheat yields are shown as grey circles and locally weighted scatterplot smoothing (loess) curve as a red line. Green labels indicate examples of years with anomalously high yields; brown labels indicate examples of years with anomalously low yields. **(b)** Same as (a) for three main wheat-growing regions (data only available for 1990-2020 at regional scale). **(c)** Anomalies of wheat yields computed by subtracting the Loess moving mean from the annual values. **(d)** Map of the three wheat-growing regions. Green indicates North Eastern Scotland, Eastern Scotland, and the North East English region (SNE); blue indicates East Midlands, Yorkshire and the Humber regions (EMYH); red indicates the South East and Eastern regions (SEE).

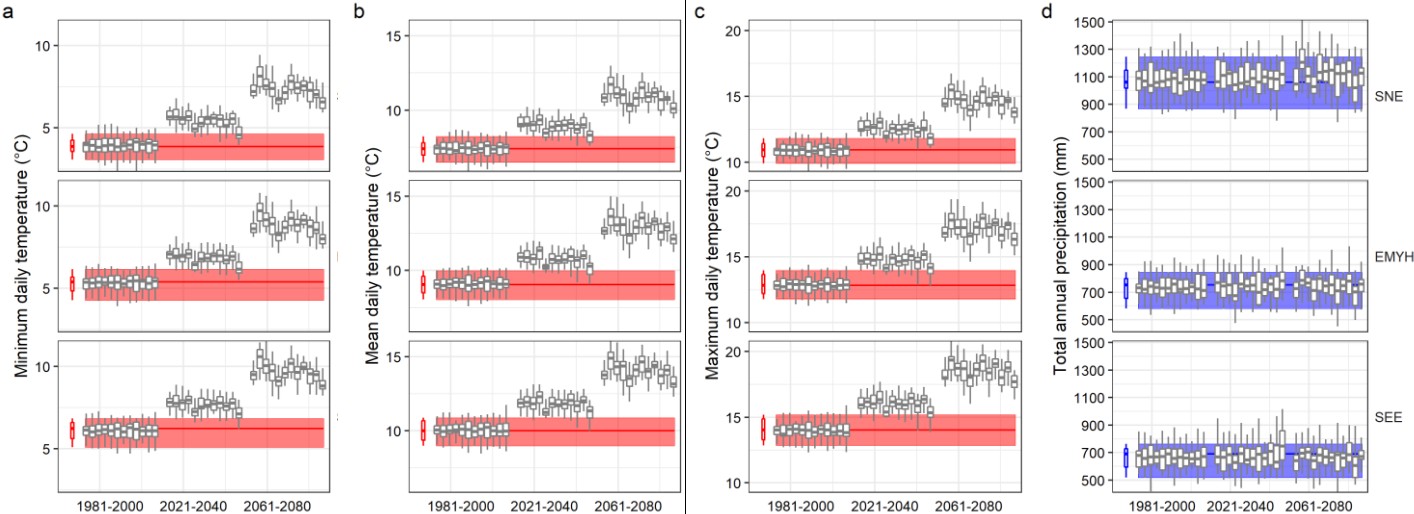

**Figure 2: Bias-correction of each UKCP 2.2km ensemble member, for (a-c) the minimum, mean and maximum daily temperature (*mean_minT*, *mean_meanT* and *mean_maxT*), respectively; and (d) total precipitation (*total_P*), in each year, for each of the three regions (top row: SNE, middle row: EMYH, bottom row: SEE). Red (blue) boxplots and rectangles indicate the range of observed temperature (precipitation) over the first period (1981-2000), based on the HadUK dataset. Grey boxplots indicate projections (one for each of the 12 UKCP Local ensembles) for three periods (historical – 1981-2000; future – 2021-2040; 2061-2080) using RCP8.5. Boxplot hinges represent 25th and 75th percentiles, and horizontal bar indicates the median. Whiskers extend to the largest value no further than 1.5 times the interquartile range (distance between 25th-75th percentiles) from the hinge.**

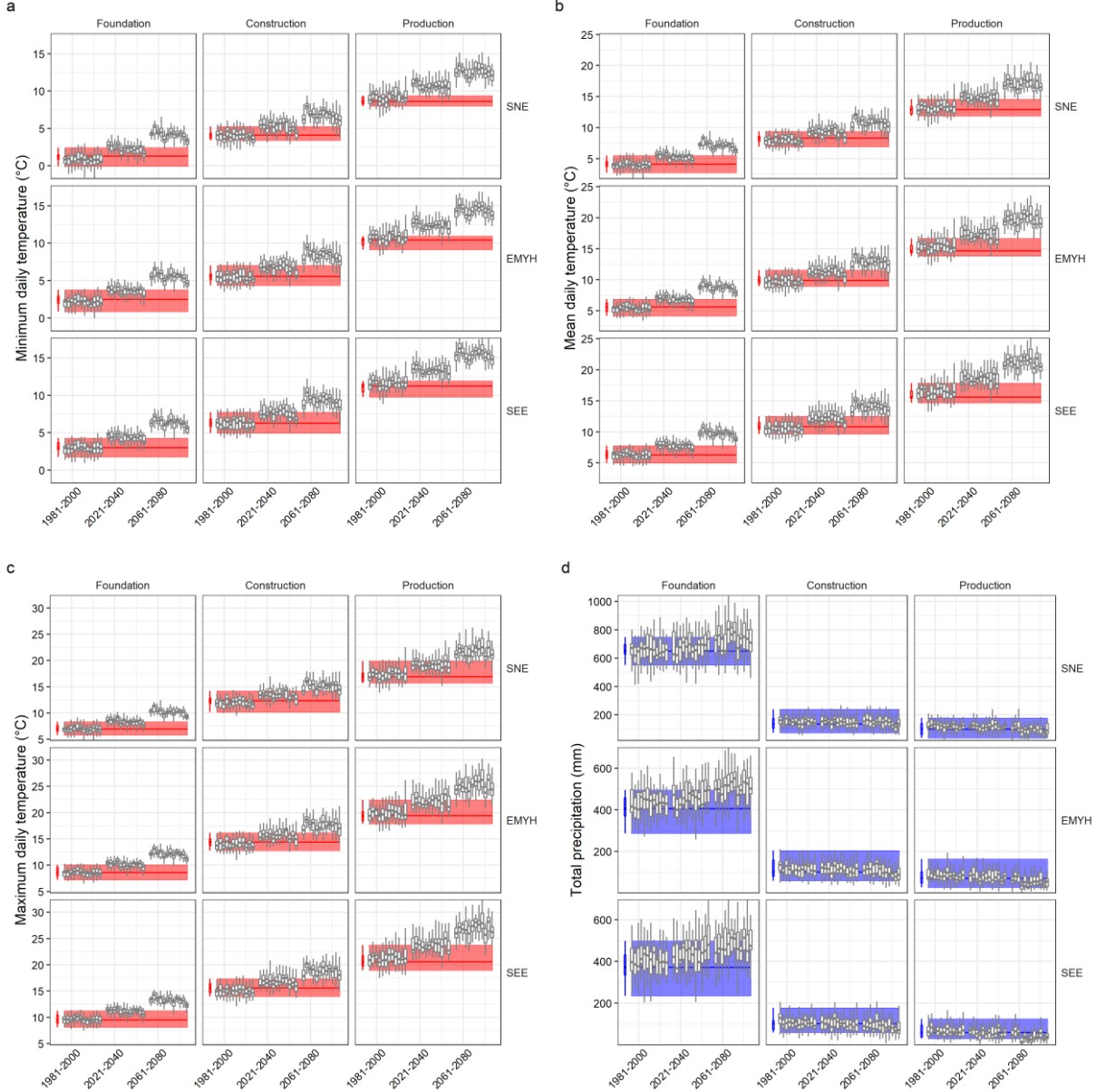

**Figure 3: Bias-correction of each UKCP 2.2km ensemble member, for (a-c) the minimum, mean and maximum daily temperature (*mean_minT*, *mean_meanT* and *mean_maxT*), respectively; and (d) total precipitation (*total_P*), within each phase, for each of the three regions (SNE, EMYH, SEE). Red (blue) boxplots and rectangles indicate the range of observed temperature (precipitation) over the first period (1981-2000), based on the HadUK dataset. Grey boxplots indicate projections (one for each of 12 UKCP Local**
**ensembles) for three periods (historical – 1981-2000; future – 2021-2040; 2061-2080) using RCP8.5. Boxplot hinges represent 25th and 75th percentiles, and horizontal bar indicates the median. Whiskers extend to the largest value no further than 1.5 times the interquartile range (distance between 25th-75th percentiles) from the hinge.**

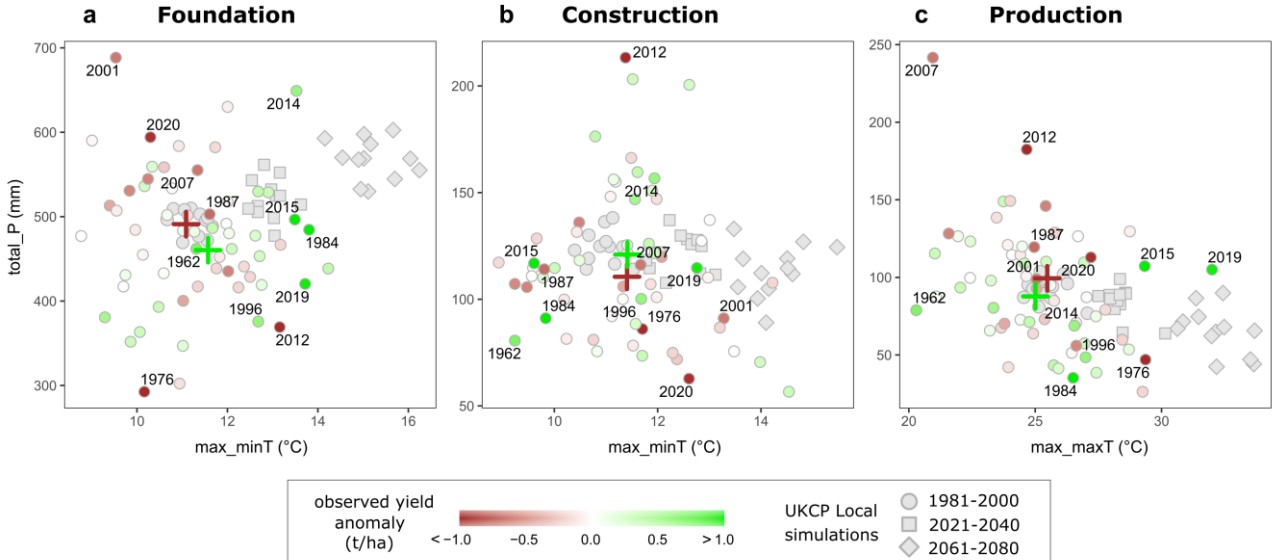

**Figure 4: Association between observed wheat yields and climate during the three wheat-growing phases. Anomalies of observed UK wheat yields are shown for total area-averaged precipitation (*total_P*) and the maximum of area-averaged minimum/maximum daily temperature within each phase (i.e. the metrics *total_P*, *max_minT*, and *max_maxT*, chosen for their associations with crop yields; Table 2), alongside UKCP projections. Columns: Foundation phase (01st October to 09th April); Construction phase (10th April to 10th June); Production phase (11th June to 26th July). Yield time series are shown for the national scale here (longer than regional time series, see Figure 1a vs 1b) and are the same in the three panels. Small green circles indicate positive yield anomalies for individual years; small brown circles indicate negative yield anomalies for individual years. Large green crosses indicate the mean for all the years with positive wheat yield anomalies; large brown crosses indicate the mean for all the years with negative wheat yield anomalies. Grey diamonds indicate UKCP Local projections of temperature and precipitation for the historical (circle: 1981-2000) and future (square: 2021-2040; diamond: 2061-2080) periods, where each symbol indicates one of the 12 ensemble members. Specific years mentioned in the main text are labelled.**

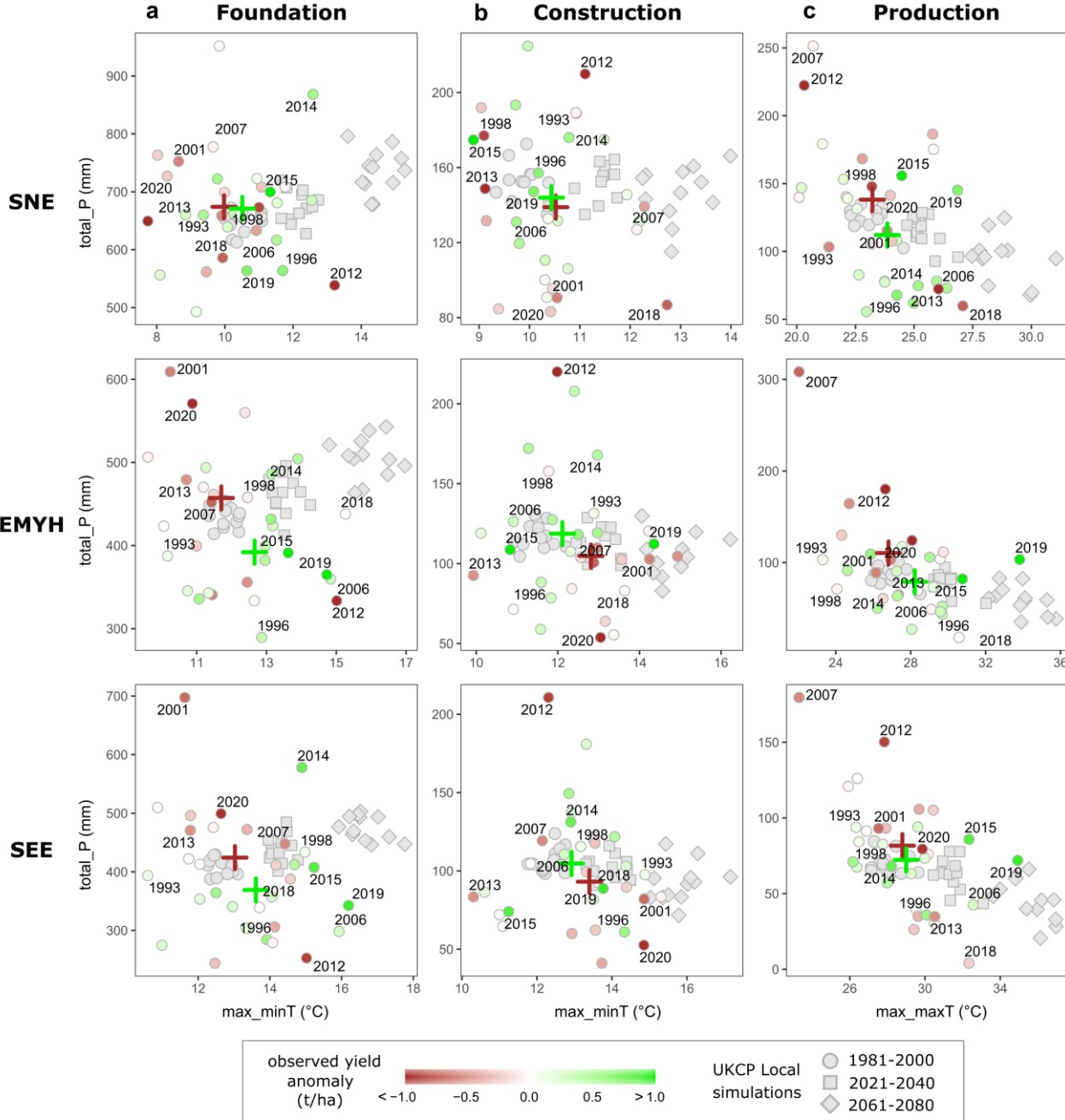

**Figure 5: Association between wheat yields and climate during the three wheat-growing phases and in each of the three UK wheat-growing regions.** Anomalies of observed UK wheat yields are shown for total area-averaged precipitation (*total_P*) and the area-averaged minimum/maximum daily temperature within each phase (i.e. the metrics *total_P*, *max_minT*, and *max_maxT*, chosen for their associations with crop yields; Table 2), alongside UKCP projections. Columns: same as Figure 4. Rows: SNE, EMYH, SEE.
Yield time series are shorter at regional scale than national (see Figure 1b). Small green circles indicate positive yield anomalies for individual years; small brown circles indicate negative yield anomalies for individual years. Symbology is the same as Figure 4.

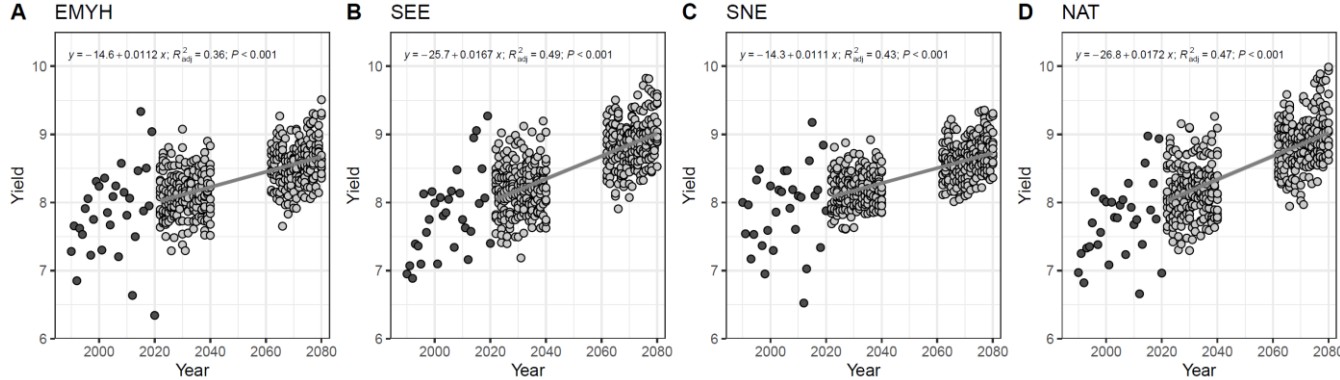

**Figure 6: Temporal trends in wheat yields (tonnes/hectare): observations and future projections. The observations (black circles, 1990-2020) are the same as in Figure 1. The projections (grey circles; 12 members per year) are obtained by forcing a multiple linear regression model (Equation 1; Table 4) obtained for each region with the UKCP Local projections of the same climate variables in each growth stage (see Methods section 2.5). Grey lines indicate the linear regressions between the ensemble of projected values and time (the regression equation, adjusted R2, and p-value are indicated on each panel).**

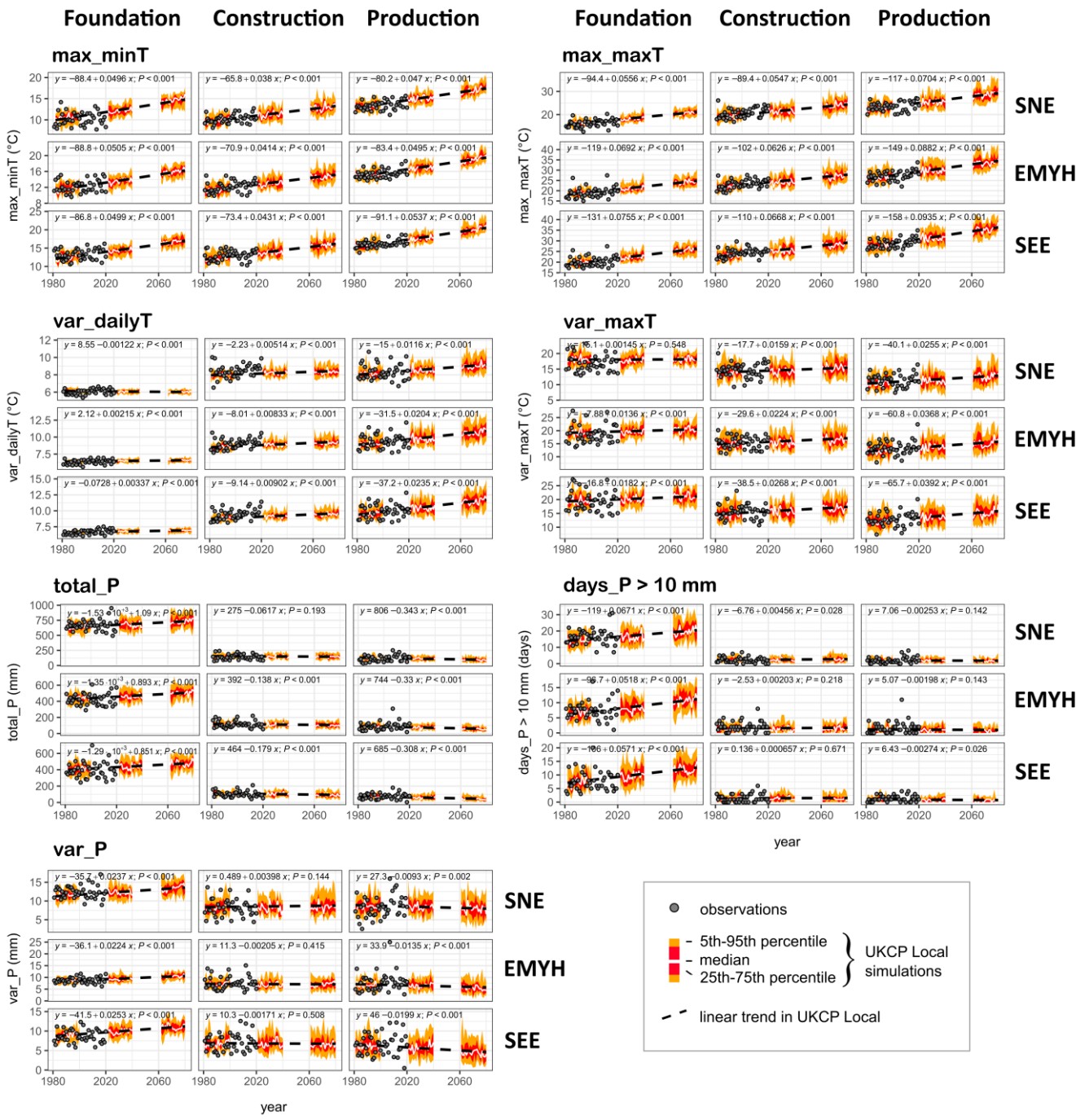

**Figure 7: Trends in key climate metrics for the three growth stages (columns) and three regions (rows). Metrics are selected from (and defined in) Table 2:** *max_minT, max_maxT, var_dailyT, var_maxT, total_P, days_P>10mm* **and** *var_P*. **Dark grey circles indicate observations; color ribbons are the 12 UKCP Local members (5th-95th percentile in orange, 25th-75th percentile in red, and the median as a white line). Linear trend lines for the UKCP simulations are shown as dashed black lines, along with the regression equation and p-value on each panel.**