# Peer review of "Resilience of UK crop yields to compound climate change"

_Earth System Dynamics, 2021_

## Author Response (AR1)

**Resilience of UK crop yields to changing climate extremes**

**Responses to the Editor and 3 Reviewers**

*In the response below, comments from the Editor and three Reviewers are copy-pasted verbatim in black font, our initial responses before revising the manuscript are in blue font, and our final responses after revisions are in red font.*

04 Apr 2022
**Editor decision: Reconsider after major revisions**

**Comments to the author:**
Dear Authors,

While all Reviewers see some merit in your submission, they recommend major revisions before it may be considered for publication in ESD. I would in particular encourage you to strengthen the description and implementation of the statistical analysis, which is key to ensuring that the conclusions drawn in your study are robust.

Best Regards,
Gabriele Messori

*We thank the Editor for his positive evaluation of our work and his invitation to revise the manuscript. We have substantially strengthened the description and implementation of the statistical analysis as suggested by the reviewers. In particular, we have added a new section, which describes more formally and completely the statistical methods used in the manuscript ("2.5 Statistical approach"). We also provide another new section, which discusses the assumptions and limitations of the work ("2.6 Assumptions and limitations"). Notable is that we have added a new multiple regression model, as suggested by one of the reviewers. This approach is similar to our previous "combined climate score" but provides a clearer picture of how crop yields might respond future to climate change under the RCP8.5 scenario. We have addressed each of the comments made by the reviewers. Our initial responses before revisions are indicated below in blue font, and final responses after revisions are indicated in red.*

**Response to Reviewer Comment 1 (RC1)**:
'Comment on esd-2021-92', Anonymous Referee #1, 05 Jan 2022

**Summary**

The submitted manuscript describes a study of the historic link between extreme weather conditions and wheat yields in the United Kingdom (UK), extended with an analysis of relevant future weather changes. It is shown that mean yields have systematically increased over time, which can be explained by technological advances, but interestingly year-to-year variability of yield has also increased in recent decades (Fig 1c). The authors then set out an analysis in which correlations between various (extreme) weather indices for temperature and precipitation, across three important wheat growing stages, and end-of-season yield are investigated. They develop a simple scoring metric that describes some of the cumulative effect of weather conditions on yield. This analysis is extended by very high resolution climate model-based projections of future temperature and precipitation conditions under a high-emission scenario, and describes future weather conditions for wheat growing.

*We thank the reviewer for their thoughtful comments on our manuscript, which will help us strengthen the research. We are glad the reviewer finds the increase in yield variability interesting and agrees that the use of wheat growing stages is important. We address each of their comments and suggestions point-by-point in blue italic font below.*

**General remarks**

**RC1.1** Whilst I appreciate that the topic is of large societal relevance, I struggle to see where the manuscript answers its research questions or if promises delivered. The link between weather and agricultural impacts is highly complex and non-linear. The authors acknowledge this in their introduction, and set out on logical path of addressing the topic and ultimately provide specific wheat sector-relevant climate projections. I fear however that the relationships are of such highly complex nature, that the present analysis does not provide satisfactory answers to the questions posed.

*We apologise if the reviewer feels our main findings were not sufficiently well highlighted. We think the issue is largely presentational. Our manuscript's key questions and conclusions are as follows:*

*Q1. Do statistically significant associations exist between observed temperature/precipitation metrics and historical wheat yields during the three crop growth stages, in the three main wheat-growing regions of the UK?*

- *In the Foundation stage, when the crop is germinating and growing slowly, there is a significant negative association between crop yields and the number of heavy rainfall days in the EMYH region (days_P>10mm). We also find significant positive associations between yield and max_minT at the national scale and in the EMYH region, and with min_meanT and min_minT in the SEE region, suggesting that yields benefit from warming temperatures (less frost).*

- *In the Construction phase, when the crop is green and growing rapidly, we find no significant associations between climate characteristics and crop yields.*

- *In the Production phase, from post-flowering to harvest, we find a consistently negative association between heavy rainfall (both total_P and days_P>10mm) and crop yield in all three regions. For total_P the association is significant in EMYH and at the national scale, and for days_P>10mm in EMYH. In contrast, good wheat yields are associated with warm summer temperatures, which can be seen in the positive associations with max_maxT or max_meanT, which are significant both nationally and in EMYH.*

- *We find cumulative climate impacts across individual growth stages, with a significant positive association between the combined climate score and wheat yields for EMYH, SEE, and the national scale, but not SNE, where the association is weaker.*

*Q2. To what extent might projections of future temperature and precipitation extremes under a high-emissions scenario impact future crop yields?*

- *Future high-temperature conditions generally fall beyond the bounds of annual variability experienced in the contemporary period for all three wheat-growing regions.*

- *For total annual precipitation, the projections do not indicate a visible increase or decrease in any of the three regions relative to the historical period; however, changes are significant when considering precipitation projections by wheat growth stages.*

- *In the Foundation phase, all regions can expect to see progressively warmer, wetter conditions in the coming decades, with significant increases in max_minT, max_maxT, total_P, and days_P>10mm. Warmer winter night temperatures are likely to prove*

*beneficial in decreasing the risk of frost damage, but concurrent increases in heavy rain may be detrimental to wheat yields.*

- *In the Construction phase, the projections indicate significant decreases in total_P in EMYH and SEE, but not SNE. There are no evident changes in heavy rain (days_P>10mm).*

- *In the Production phase, we find much warmer and somewhat drier conditions in all three regions, with significant increases in max_minT, max_maxT, and equally in temperature variability (var_dailyT and var_maxT). Warmer and drier mean conditions may enhance wheat yields, but increases in high temperatures (outside the range experienced in the historic period) and heat variability may increase plant stress, while the significant decreases in rainfall (total_P) may threaten adequate water supply.*

- *Projections of future temperature and precipitation conditions do not significantly aggravate our simple combined climate score (relying on max_minT, max_maxT and total_P), suggesting the beneficial impacts of warming conditions (e.g. reduced frost risk) may be offset by significant increases in heavy rainfall in the Foundation phase and enhanced drought conditions in the Production phase.*

*We purposely adopted a nuanced approach in answering these questions as we believe it is important to highlight that this is a complex topic (as the reviewer points out), without seeking to over-interpret the findings. We will revise the text to ensure that the findings are clearly laid out, emphasising where associations are significant (as above). We will also explain any assumptions and limitations, as discussed in our reply to RC1.10.*

*The two research questions (Q1 and Q2 above) can be found at the end of the Introduction (no change to the text). Our main changes are that the Results and Conclusions (sections 3-4) have been carefully updated, based on the revised manuscript and the results of the new regression model. We have also modified the Conclusions so they clearly answer the two research questions.*

**RC1.2** The first research aim (finding statistical associations) results in Table 2. Only one of three regions shows any stat.significant relations (at p=0.05 level) between the weather indices and yield.

*A1.2 As the reviewer rightly points out, the strength of the associations between climate metrics and crop yields varies depending on the growth phase and region. For transparency, we show all the associations and their relative strength. We feel it is just as important to show where these associations are non-significant as where they are significant, based on the historical data.*

*The correlations table has been retained in the revised manuscript because we feel that the weakness of some statistical associations is also an important finding to present. With this in mind, we have worked carefully through the revised manuscript, as well as the Conclusions, to ensure that we emphasize which of the associations are significant and which are not.*

**RC1.3** The accompanying text leans very heavily on anecdotal evidence, which I fear may lean towards overinterpretation of single events.

*A1.3 We believe that descriptions of some individual events are important: many of the answers to our questions are understood by farmers and are present in the grey literature, but not in the academic literature. Without discussion of important historical events, the manuscript would be very dry. However, we entirely agree with the reviewer that individual events should only be used to illustrate significant associations, and we will carefully edit the text to make sure this is the case (i.e., in cases where the associations are not significant, we will say so explicitly).*

*The manuscript text has been updated carefully in this context, so that the individual events are only used to illustrate significant associations, or provide relevant information. In cases where associations are not significant, this has been clarified.*

**RC1.4** To account for temporally compounding effects a simple scoring metric is developed (Fig 6), failing to take into account developments in agricultural science. Why haven't the authors followed their own advise (last sentence) and employed process-based crop models or AI methods to find robust relationships between weather and yield?

**A1.4** *To be effective, process-based crop or AI-driven models of actual crop yields require parameterisation and calibration with large volumes of data on local conditions and agricultural management. Obtaining these data over large spatial extents remains challenging. Our approach circumvents these issues by trying to develop a generalisable approach to understand if there are associations in the observational data. Our final sentence is thus an ambition for future research rather than an alternative method for our current analysis.*

*We have clarified the manuscript conclusions to explain that further data is required to develop such approaches, including creating robust agricultural models. To get us closer to what we might expect such models to explain when available, we have replaced our 'combined climate score' with a multiple regression, which allows us to provide more nuanced data-driven projections of future yields, using the UKCP Local simulations.*

**RC1.5** Then, the future climate projections are based on a single climate model. Though high-resolution modelling without doubt adds value, a multi-model perspective is needed to provide 'reliable' projections. I advise the authors to add a comparison of projections in CMIP6 or CORDEX to UKCP, such that readers may get a feeling of where these projections lie within the larger model-related uncertainty.

*Some information from CMIP models is included through the comparison with UKCP Probabilistic projections. UKCP Local projections are generally within the 5-95% probability levels of the UKCP Probabilistic projections, which include multi-model information from CMIP5 (see reply RC1.22 below for details). We are sorry this was not as clear as it could have been.*

*The future UKCP Local climate projections are based on an ensemble of 12 convection-permitting (2.2km grid spacing) simulations, which sample uncertainty in the physics of the driving model. In particular, uncertain parameters within the model physics are perturbed within acceptable bounds, allowing a first estimate of uncertainty in future changes. This uncertainty however is likely to be underestimated, since UKCP Local only downscales perturbed physics versions of the Hadley Centre Climate Model. This is being addressed in new simulations, which are underway as part of an update to UKCP18, downscaling selected CMIP5 models to 2.2km resolution over the UK. These new simulations will augment UKCP Local, sampling a wider range of uncertainties. Unfortunately these results are not available currently and so cannot be included in this paper, but will be exploited in future analysis. The current analysis presented in this paper nevertheless is a major step forward in using state-of-the-art high-resolution climate projections that provide reliable information on changes in local weather extremes. In the revised manuscript, we will provide a clear description of how the UKCP projections compare with the larger model-related uncertainty.*

*We agree a discussion of the climate modelling assumptions and limitations is important. We have added further explanation of the perturbed physics ensemble (PPE) aspects of the UKCP local simulations in section 2.6 ('Assumptions and limitations'). In that section we explain how the 12 members of the high-resolution ensemble describe both internal climate variability and the climate modelling uncertainty in the driving model. We discuss how the UKCP Local simulations compare with the CMIP simulations, and explain how the trends of the UKCP Local*

*simulations at least partially cover the range of uncertainty and trends that occur in the ESMs developed by other climate research centres (see section, 2.6, point 6).*

**RC1.6** But more importantly, rather than analysing how crops respond to future weather conditions (which is what the title of this manuscript implies), changing weather conditions in three growing stages are discussed. Given the limited relationships between weather and yield that were found, does this really provide useful information for the UK agricultural sector?

*The title was chosen to encompass changing weather conditions in both the past and future ("Resilience of UK crop yields to changing climate extremes"). We respectfully disagree with the reviewer here, as the manuscript provides clear conclusions, which have implications for the UK agricultural sector. We refer to the detailed summary of conclusions in response to RC1.1.*

*The manuscript abstract, methods, discussion and conclusions sections have all been updated. A multiple regression model has been developed using observations of both crop yields and climate metrics over the historical period. Projections of future crop yields have been obtained by driving the regression model with the UKCP Local simulations. Therefore the manuscript does analyse how crops might respond to future weather conditions under the RCP8.5 scenario (with assumptions and limitations of the approach discussed in the new section 2.6).*

**Major points**

**RC1.7** Section 2.4 - In table 3 I see you have bias-corrected each ensemble member separately. Though I haven't worked with perturbed physics large ensembles, the normal procedure for bias correcting would be to do a single bias correction for all ensemble members, as differences are due to internal variability not indicative of different mean climates. Are the imposed physics changes so large that this is different in this case?

*If we were using an ensemble with the same physical structure then the reviewer is correct: we would apply a single bias correction. In the case of a perturbed physics ensemble, however, each ensemble member is typically regarded as a different model, and so one bias correction is applied for each ensemble member. Here, the physics of the driving model have been perturbed. Thus, differences between the UKCP Local ensemble members are due to differences in the physics of the governing forcing ESM (which justifies a separate bias correction for each ensemble member) as well as natural variability. We will clarify and explain this point within the revised manuscript.*

*We have clarified why we use ensemble member-specific bias correction in the manuscript. Please see our enhanced Section 2.4 on Bias correction, as described above.*

**RC1.8** Section 3.2 - The relationships noted in this section are very anecdotical. Table 2 provides the quantitative correlation coefficients, which are only in a few cases statistically significant. Please add more quantitative evidence of the suggested relations, or note in the text that despite anecdotical evidence there is no statistical link. I think it is easy to overfit/overanalyse seemingly simple relations (e.g. wet conditions lead to low yield), when in reality the interactions between plant and weather is very complex and highly non-linear.

*We agree with the reviewer that interactions between plants and weather are generally complex and nonlinear. We will carefully revise the manuscript to make sure that it is clear which associations are not significant. We will also make sure that we are not over-extrapolating from individual events.*

*The manuscript text has been revised: we use formal statistics before introducing specific examples (years), but we still look at specific years, because they provide relevant insight.*

**RC1.9** Line 295, line 318 - I hadn't noted evidence in the results section supporting the conclusion that increased inter annual yield variability is linked to "one-to-one correlations with temperature or precipitation extremes" or "the recent increase in yield volatility is associated with combined climate metrics". Please either emphasise this more in the results section, or remove from the conclusions.

*We will revise the text and revisit our results to make sure this point is either better explained or removed from the conclusions.*

*The manuscript text has been revised in both places to clarify how we assess the associations between the crop yields and climate metrics.*

**RC1.10** I miss a discussion of the assumptions that went into this work, and how these assumptions might influence the results. One item would be the use of fixed-in-time growth stages, in reality these are weather dependent, and plant vulnerabilities to extreme temperatures or precipitation can thus be different from one July to another July (for example).

*Thank you for pointing this out; we will add a discussion of assumptions including the use of fixed-in-time growth stages. In the manuscript, we did not use the detailed (99) physiological growth stages (AHDB 2022), but rather the high-level growth stages (which are defined over long time periods) to split the year into key stages of wheat growth, and to mitigate against weather dependencies. Other assumptions that are worth describing include the assumption that all wheat varieties behave similarly (both in the past and future). However, one of the advantages of our empirical data-driven approach is that there are fewer assumptions than in a process-based model approach.*

*AHDB 2022, The growth stages of cereals, Agriculture and Horticulture Development Board, Kenilworth, Warwickshire, https://ahdb.org.uk/knowledge-library/the-growth-stages-of-cereals*

*We agree this is a very relevant point. We have included a whole new section on the assumptions that went into the work (Section 2.6 "Assumptions and Limitations"), which includes a discussion of the point mentioned by the Reviewer regarding the use of fixed-in-time growth stages.*

**Minor points**

**RC1.11** Fig 1 caption - I very much dislike the bracket-way of scientific writing. There is no word limit in ESD, I strongly encourage the authors to rewrite their statement in two sentences. "Green (brown) labels indicate examples of years with anomalously high 485 (low) yields."

*We are happy to make the change.*

*Done (here and elsewhere in the manuscript).*

**RC1.12** Line 50 - Indeed there is a lot of climate research into weather extremes, but there is a vast quantity of climate impact research as well in how these will influence for example crops. A slight rearrangement of this sentence is asked for. And maybe a few more recent examples from the literature (e.g. Ben-Ari et al 2018).

*We will rearrange the sentence and add further examples from climate impact research and the crop literature (including the recommended paper).*

*Done.*

**RC1.13** Line 85 - Please clarify "any incomplete crop growth stages", does this relate to gaps in the PR/TAS data, or too fast progression through the stage? If the latter, would that not be caused by climate extremes?

*This relates to gaps in the data (incomplete seasons were removed). We will clarify this point in the text.*

*Done.*

**RC1.14** Line 104 - A model realistically simulating present-day climate is a requisite for making a seemingly reliable projection, but it is not guaranteed of course. The response to GHG forcing can still be very wrong. And since only RCP8.5 is used as forcing, further doubt on the 'credibility' of the projections is added (see e.g. Hausfather and Peters et al 2020). This does not discredit your analysis, but does put some limitations on the 'credibility of projections' made. Please add a sentence noting these issues (single model, single GHG scenario), maybe use the first paragraph of section 3.4 and remove it there.

*The reviewer raises an important point. Although we did already mention the use of a single model, single GHG scenario, we will discuss these aspects further within an explicit 'assumptions and limitations' section. We will also better explain the benefits of using a perturbed physics ensemble.*

*We have moved the first paragraph of section 3.4 to the new "Assumptions and limitations" section (section 2.6) as suggested by the Reviewer. We have also mentioned that although the analysis relies on one model, the members of our UKCP local climate projections are driven by a perturbed physics ensemble (PPE), and so samples a substantial amount of uncertainty in future climate evolution. We have now explained more clearly the strengths and limitations of using a perturbed physics ensemble.*

**RC1.15** Fig 2 - I originally thought the panels a-c showed the UK as a whole, only noting later that maybe the small region labels on the right count for all panels. Maybe add these inside the plot, or explicitly state this in the caption.

*Thank you. We will make the labels more visible on the plot, and better phrase the caption.*

*We have mentioned this in the caption, to avoid over-cluttering the figure.*

**RC1.16** Line 154 - I don't understand why you would think those are related, on the one hand growth due to technology and on the other hand increased variance? I'd say for the first you have a very good argument, and the second is an interesting question indeed, with the link to increasing weather extremes as a good hypothesis.

*This is a misunderstanding; our aim was not to imply that growth and variance were related. We will ensure the text is rephrased for clarity.*

*We have rephrased the sentence for clarity.*

**RC1.17** Line 169, Fig 4/5 - These figures show a lot of data in a small panel. As you don't discuss any correlation between PR and TAS, why would you show them in this way? Wouldn't a 'simple' scatter plot between PR and yield better show this conclusion? Furthermore, yellow yield years show an extreme PR in one of the seasons, but how many normal/high yield years do the same? E.g. from Table2 only EMYH in the production phase shows a statistically significant correlation between PR and yield. Then, what do the grey crosses add to the figure?

*The plots are not intended to show correlations but instead to show the spread of individual years and their associated yields throughout the climatic 'space' generated by the interaction*

*between PR and TAS. The projections (crosses) then allow the reader to see how future seasons may compare with present seasons in terms of warming/wetting.*

*We have removed the crosses from the projections (standard deviations). This now makes the figures clearer and simpler to read. We also removed the ellipses and instead show the mean point associated with positive and negative yield anomalies (crosses). We believe this makes the diagram clearer, while retaining the overall message.*

**RC1.18** Table 2 - the horizontal line separating TAS and PR measures is one row too low.

*Thank you for pointing this out - we will make the change.*

*This is a formatting issue caused by the Word-to-PDF conversion. It should look okay now.*

**RC1.19** Line 205 - Add here that the growing phases in real plants are determined by their growth, rather than calendar days. So a phase can last longer, to have the desired number of growing degree days for example, delaying the crop, but resulting in the expected yield. The calendar-fixed phases are a simplification of this process.

*This is a good point; we will edit the text accordingly, thank you.*

*The text has been added to the revised manuscript (section 3.3). It is also discussed in the new section on Assumptions and limitations (section 2.6).*

**RC1.20** Fig 6 - A few remarks: (i) Please separate the projections from the observed data, maybe in a second row of figures below the first one. (ii) I don't understand where the future yield data are coming from? The relationship of black circles and triangles and grey shading is surprisingly (doubtfully?) linear, and fully captures the eye of the reader. The dots show very much variability, by eye alone I doubt one would have been able to draw the correct regression line through them. (iii) please add fitted regression lines using observed data (I assume the statistics plotted are those lines), and maybe for the national subplot also show data from before 1990.

*(i-ii) The reviewer's comments are helpful and reveal that we should have better explained the figure. We will clarify this in the caption and main text. By overlaying the observations and projections of the climate score, it is easier to compare them. (iii) We will make the regression lines more visible, and add the older data for the national subplot as suggested.*

*After implementing the proposed multiple linear regression model, we decided this provided the best approach, and so removed the parallel combined climate score figure. Hence we provide the multiple regression model-based projections in Figure 6, which provide similar information to the original multiple regression but with greater nuance.*

*For the national scale panel (Figure 6D), we initially followed the reviewer's advice and included a longer period of observed data (shown below). However, the association between climate and yields in the 1960s-1980s is less representative of the association between climate and yields of today (due to different agricultural practices and lower yields). Including all the historical data is thus less appropriate (see figure below, not included in the revised manuscript). Therefore, we preferred to keep the same period (1990-2020) as for the regional models as this makes the results more consistent and easier to compare with the correlations provided Table 2.*

[Figure]

**RC1.21** Section 3.4 - The section title is misleading, general forced climatic changes are discussed, not crop-specific climatic changes.

*This is a good point; we will make the change, thank you.*

*The title of section 3.4. has been revised to "Annual projections of future climate conditions and implications for crop yields".*

**RC1.22** Line 245 - I imagine the UKCP lie in the upper/lower-percentiles of the full CMIP5/6 ensemble, but not fully outside? "UKCP simulations tend to sample greater future warming and drying in summer compared to the full…"

*This is a good question; we will add a more detailed discussion in the manuscript of how the UKCP compare to the CMIP ensemble. The figure below from Kendon et al. (2021) shows that UKCP Local projections (olive green dots) are generally within the 5-95% probability levels of the UKCP Probabilistic projections (black boxplots, which include some multi-model information from CMIP5). One exception is winter when the UKCP Local show some precipitation responses above the 95% level. This is understood and relates to the improved representation of winter-time convective showers in the Local 2.2km model (Kendon et al 2020). UKCP Local projections sample relatively high temperature changes, with few outcomes cooler than the median of the UKCP Probabilistic projections in winter and none in summer. Changes in summer precipitation show a considerable drying in the Local projections (2.2km), whereas the 13 CMIP5 simulations and the UKCP Probabilistic projections indicate that outcomes with more modest reductions or small increases should also be considered.*

[Figure]

*Above: Figure 5.1 from Kendon et al. 2021. Comparison of seasonal mean changes across UKCP18 products. Projected changes for 2061-2080 relative to 1981-2000 for Scotland and England in (top) JJA and (bottom) DJF, under RCP8.5 emissions. Results are shown for surface air temperature (left, °C) and precipitation (right, %). Box and whiskers denote the 5, 10, 25, 50, 75, 90 and 95% probability levels of the UKCP probabilistic projections (Strand 1). Orange dots (with STD in red) denote members of GC3.05-PPE and blue dots those of CMIP5-13, which together comprise the UKCP Global (60km) projections (Strand 2). Pink dots (with STD in purple) show the Regional (12km) projections and green dots (with STD in dark green) those of the Local (2.2km) projections (with the original convection permitting model in fluoro-green and the new convection permitting model (used in this manuscript) in olive-green, Strand 3).*

*References*

*Kendon E J et al (2021) Update to UKCP Local (2.2km) projections, July 2021, Met Office, Exeter, UK https://www.metoffice.gov.uk/pub/data/weather/uk/ukcp18/science-reports/ukcp18_local_update_report_2021.pdf*

*Kendon, E. J., et al (2020) Greater future UK winter precipitation increase in new convection-permitting scenarios. J Climate. DOI: 10.1175/JCLI-D-20-0089.1*

*We have added a discussion of how the UKCP Local simulations compare to the full CMIP ensemble, again noting that the ensemble members we use are driven by a perturbed physics ensemble (PPE). These perturbations are designed so that the UKCP Local simulations at least partially cover the range of uncertainty and trends that occur in the CMIP ESM archive. We discuss this point in the new section 2.6 on modelling assumptions and limitations. These particular changes were also requested by another of our reviewers.*

**RC1.23** Fig 7 - I'm not sure this is the best way of showing the data. 10 lines on top of each other, plus the ensemble mean, and then for each the regression line. I had to zoom in to 500% to read the data. Maybe consider only showing with shading the min-P25-P50-P75-max

across the ensemble, and the ensemble mean regression line? This would show less data, but I think more information.

*We will make the change as suggested by the reviewer.*

*The figure has been updated accordingly, and we agree that it is now easier to read.*

*Thank you for the helpful review, which helped us strengthen and clarify the work.*

**Response to Reviewer Comment 2 (RC2):**
'Comment on esd-2021-92', Corey Lesk, 13 Jan 2022

This paper seeks to understand historical links between climate and variability in UK wheat yields, and examine the implications of future climate change as projected using a convection-resolving climate model. It considers differential yield responses across growth stages, and tries to then aggregate these stages to assess compensating or amplifying impacts. This latter aspect is the main novelty of this research, which I think is useful and timely. I also appreciate the effort to consider compensating effects between growth stages and between heat and water via this aggregate climate scoring approach, and in Figures 4-5. There is increasing attention to these joint affects, so this paper has the potential to add some clarity here as well.

*We would like to thank Corey Lesk for his very helpful review of our manuscript. We are pleased he finds the two main aspects novel and useful – i.e. assessing compensating/amplifying impacts across different growth stages, and considering heat and water extremes together. We are also pleased that the paper is seen as "timely". We address each of the comments and suggestions point-by-point in blue italic font below.*

**RC2.1** I have two main critiques that should be addressed. First, the statistical analysis is not adequately described, and based on what I can surmise from the sparse detail, it is probably not the strongest approach. Second, the assessment of future impacts is only driven by data on the climate side, and the crop impacts are only qualitatively discussed. This sells the historical climate-yield relationships short: why not use your historical results for a data-driven estimate of future impacts? Further, your results and other research show how multivariate climate variation/change could lead to compensating or compounding impacts on crops, the potential for which could be more robustly and objectively assessed through a more quantitative approach.

*We appreciate this comment. First, we will make sure that our statistical analysis is fully described in the revised manuscript, and we will test the additional statistical approach suggested below in RC2.2 (a multivariate statistical model). Second, regarding the assessment of future climate impacts on crop yields, it is true that a multivariate statistical model using the historical observations could then be driven with the UKCP projections. We will explore this data-driven approach as suggested (e.g., temperature and precipitation variables for each growth stage all included in one yield model). We discuss this in more detail in response to the next comment.*

*First, a new section has been added describing the statistical approach (section 2.5 "Statistical approach"). Second, a new multiple linear regression model was developed using the historical observations. This model was then driven using the UKCP Local climate projections to obtain a 'data-driven estimate of future impacts', as suggested. This approach is not dissimilar to our previous 'combined climate score' but the method is more conventional and so the results are easier to interpret. This approach is described in the methods section (section 2.5) and the results (and Figure 6) have been updated accordingly.*

**RC2.2.1** On statistical analysis: The methods is missing any description of the statistical analysis, justification for model specification, etc. This makes it fairly hard to assess the reliability of the results, and what they mean. I gather that the analysis is pairwise two-variable Pearson correlations (yield vs. each climate variable). The authors then use these results to develop a scoring system to combine variables/growth stages, which is not necessarily a bad approach, and the results in Figure 6 seem pretty strong. But this is not a widely used approach, and given lack of detailed methods, it is hard to assess. **RC2.2.2** Rather, multivariate regression (i.e., temperature and precipitation variables for each growth stage all included in one yield model) is what is typical. There are both benefits and pitfalls to it, but it would improve confidence to try this more widely-vetted method and see if results are consistent, and would enable a more self-consistent way to assess compensations. **RC2.2.3** Further, this multivariate regression approach is more suited to then actually projecting yield based on multivariate projections from climate models. You may also consider non-linear yield responses. **RC2.2.4** Finally, only p-values are mentioned in the text, which only provide limited information. I see Pearson coefficients in a table, but their relative magnitudes are not discussed. And the effect size (i.e. slope coefficient) underlying these correlations also provide useful information (steepness of yield response to climate variable), so may be helpful to discuss.

*AC2.2.1 Yes, the statistical analysis in Table 2 was simply pairwise two-variable Pearson correlations (yield vs. each climate variable), as indicated in the caption. We will ensure the statistical analysis is fully described in the revised manuscript.*

*AC2.2.2 The Reviewer makes a very good suggestion about developing a multivariate regression model with the historical observations, and we will test this approach.*

*AC2.2.3 As suggested, we will use the same model to project future yield using the climate model projections. We will explore this approach in the revised manuscript.*

*AC2.2.4 Yes, we agree here too. Whether we keep the existing statistical analysis in the manuscript, or enhance it, we will ensure that it is thoroughly described and reproducible to a reader. We will include further statistical diagnostics, including the relative magnitudes of correlation coefficients and slope coefficients.*

*AC2.2.1 A new section has been added describing the statistical approach (section 2.5).*

*AC2.2.2 A multiple regression model has been developed using the historical observations and we agree this model provides greater nuance than the previous climate scoring approach.*

*AC2.2.3 We employed this new model to project future crop yields*

*AC2.2.4 We provided the coefficients of the new model in a new table (Table 4), which can be easily interpreted by others. The regression equations and p-values are now also provided on Figures 6-7.*

**RC2.3** Another methodological issue is reliance on interpreting specific years relative to statistical results, which often lead the paragraphs in the results. I actually really like this for its concreteness, but it is not a super robust method and seems prone to cherry-picking years that fit the narrative. I think this can be remedied by trying to frame these claims more as discussion points and reducing their prominence in the results. Alternatively, you could formalize your method for selecting key years, and describe it in the text.

*This is a fair point, which was also raised by Reviewer 1. We will ensure the revised manuscript will have this alternative ordering, i.e. that the descriptions of individual years do not lead our results and instead are used more as discussion points, with less prominence in the results.*

*The manuscript text has been entirely revised in this context. We still continue to use example years to illustrate specific associations. However, we clearly indicate which of the associations are statistically significant.*

**RC2.4.1** Another important limitation of this research is its use of only one climate model under only one climate forcing scenario. This leaves important uncertainties in emissions trajectories and climate responses unquantified.

*AC2.4.1 We understand the concern regarding the use of one climate model. However, the driving Earth System Model of UKCP Local is subjected to a range of plausible parameter variation (perturbed physics experiments). Hence the different ensemble members at least partially represent the range of uncertainty in climate models held in the CMIP ensembles (see last bullet point of page 5 of this document: https://www.metoffice.gov.uk/binaries/ content/assets/metofficegovuk/pdf/research/ukcp/ukcp18-factsheet-local-2.2km.pdf). We acknowledge that we are not sampling other international climate models, and so likely underestimate climate modelling uncertainty (i.e. only sampling a small part of the range). This could perhaps be addressed in future work once CMIP5 driven UKCP Local projections become available. Additionally, as discussed in response to RC1.22, the UKCP Local projections are generally within the 5-95% probability levels of the UKCP Probabilistic projections, which include some multi-model information from CMIP5. We will more fully justify and explain why UKCP is particularly useful for the UK (e.g. it gives reduced biases in both summer and winter mean rainfall). We will also reiterate that the process representation of rainfall-based effects in UKCP Local are considered 'state-of-the-art', disaggregating large-scale changes accurately, as possible with a convection-permitting model.*

This comment is similar to others raised, including by other reviewers, and has been carefully addressed. In the revised manuscript we explain that the UKCP Local simulations are driven by a perturbed physics ensemble (PPE) of a single forcing Earth System Model (ESM). They at least partially cover the range of uncertainty and trends that occur in the ESMs developed by other climate research centres. We have included a full discussion of this point (one climate model, but perturbed physics), and we acknowledge the limitation of one forcing scenario. These explanations are in the new section 2.6 ("Assumptions and limitations").

**RC2.4.2** The RCP8.5 scenario also is falling out of favor in some circles, as it assumes implausibly high emissions – the authors acknowledge this late in the paper, but don't strongly justify why we should nevertheless be focusing on an unlikely future. It would probably be useful to include RCP2.6 or 4.5, or at very least acknowledge that the paper doesn't address emissions uncertainty.

*AC2.4.2 We appreciate that the RCP8.5 scenario may not be the most likely scenario, but we will more clearly explain why it is used in our study. The first reason is that this is the only scenario for which UKCP Local projections were performed. RCP8.5 was deliberately chosen as the configuration for UKCP Local simulations to maximise the signal to noise, while still representing a plausible scenario. Using a high emissions scenario has the advantage that we can infer changes for other lower emissions scenarios using scaling approaches. We will clarify that the paper does not address emissions uncertainty, although a reasonable assumption, to first order, is that changes under lower emissions will broadly scale with change in GHG radiative forcing.*

We have included a full discussion of our reasons for using the RCP8.5 scenario in the new manuscript section 2.6 ("Assumptions and limitations").

**RC2.4.1 (similar point as above)** The implications of using one climate model should also be justified – is the HadGEM3/HadREM3 nested model particularly useful for the region? The use of a 12 member ensemble helps, but I notice that some years (often with important yield impacts) in Figures 4-5 fall outside the whiskers of the historical model data, raising questions of whether this model can reproduce these conditions (historically or in the future). We know models have such deficiencies –using more than one can help at least partly constrain uncertainty.

*We have included a discussion of the use of UKCP Local simulations and how they compare with other climate model projections in the new section 2.6 ("Assumptions and limitations"). Please see above response AC2.4.1.*

**RC2.4.3** Small comment: impacts of rising CO2 on crop water use will be important in the future, as you mention in the intro. It's a huge uncertainty and hard to model, but should probably discuss its relevance for your projections.

*AC2.4.3 We agree that the impact of rising CO2 on crop physiological response and water use is an important uncertainty and will make sure that we discuss its relevance in the revised manuscript (e.g. Ewert et al. 2002 and other references). A key point of our manuscript is that we are likely to move outside of the climatic envelope which wheat farming in the UK has previously experienced and adapted to. Thus, the high levels of uncertainty around the effects of rising CO2 on crop growth and yield are only likely to increase the degree to which farmers may struggle to adapt to and mitigate against climate impacts.*

*Ewert, F., Rodriguez, D., Jamieson, P., Semenov, M. A., Mitchell, R. A. C., Goudriaan, J., ... & Villalobos, F. (2002). Effects of elevated CO2 and drought on wheat: testing crop simulation models for different experimental and climatic conditions. Agriculture, Ecosystems & Environment, 93(1-3), 249-266. https://doi.org/10.1016/S0167-8809(01)00352-8*

*We have mentioned this point at the end of section 3.5. While there are many research papers trying to identify the $CO_2$ effect on vegetation, for crops (and in particular wheat), key references remain Ewer et al. (2002) and Swann et al. (2016). We now cite these papers.*

**RC2.5** Finally, I think you could consider in a bit more depth the interactions between temperature and precipitation both in the climate and for crops. For instance, very hot conditions in the UK can often only be reached with a dry land surface (visible as apparent negative temp-precip correlations during production, Fig's 4-5). Miralles et al. 2019 is useful reference on these processes. Cool and wet conditions could also be linked physically, with implications for crop impacts. This raises questions about the independence of heat and moisture impacts, which is a problem here since they are only assessed one-at-a-time using Pearson's correlations (multivariate regression could help capture the interaction). Further, joint impacts of changes in temp and precip in the future could be discussed more, see line comments.

*We agree with you that the interactions between temperature and precipitation and their impacts are important and should be further considered. We will explore these interlinkages (interdependence) and impacts using the multivariate approach suggested in RC2.2 - thank you for the suggestion. We will cite the suggested reference (Miralles et al. 2019), and we will also discuss potential joint impacts of changes in temperature and precipitation into the future as GHGs rise.*

*A discussion of the interactions between temperature and precipitation, both in the climate and in terms of their implications for crops, has been added in the manuscript and at the end of section 3.5. We appreciate the suggestion to include a multiple regression model, which is a more common approach to define simultaneous variation in drivers than our original statistical structure.*

Thanks for the nice paper! I think it will be a useful publication once some issues are addressed.

*Thank you very much for your positive and helpful comments on our manuscript!*

**Line comments:**

**RC2.6** Line 36: Could cite more recent papers on this: Ray et al. 2019, Ortiz-Bobea et al. 2021

*Thank you. We will cite more recent papers, including these two that you suggest.*

*Done.*

**RC2.7** Line 55: Ainsworth and Long 2021 would be a useful reference here

*Thank you for the suggested reference; we will include it.*

*Done.*

**RC2.8** Line 56: Soil moisture, precipitation intensity/distribution ref?

*Sorry, this question isn't entirely clear but we can include a reference to the importance of soil moisture and rainfall intensity and their impacts for crop yields.*

*We now mention the benefits of rainfall intensification at this point in the manuscript.*

**RC2.9** Line 80: Could be helpful to motivate this step. Presumably, you do this to remove long-term yield trends (due to technology, climate, co2) and isolate annual anomalies relative to this.

*The Reviewer is referring to the fact we subtract this running mean from each annual value. Yes, indeed, we do this to remove the trend and isolate annual anomalies which we expect to be related to interannual climate variability rather than, say, long-term technological improvements. We will clarify this normalisation in the revised manuscript.*

*Done.*

**RC2.10** Line 100: This threshold for heavy rainfall should be justified and/or its influence should be tested. For instance, Lesk et al. 2020 found extreme rainfall impacts only at high intensities >50mm/hr for US maize and soy (how this maps to daily scale is unclear, but a 10mm/hr threshold would preclude these damaging intensities). Others have used more holistic distributional measures like the daily rainfall GINI coefficient (Shortridge 2019). I'm not aware of equivalent studies for wheat, but these could be good references to add to Zampieri et al. 2017 in line 56 to bring in studies in sub-seasonal rainfall distribution.

*Thank you, this is helpful information. We will justify the choice of threshold: e.g. how it compares with the annual rainfall distribution over the British Isles; its relevance in the context of UK crop yields; and how it compares with the thresholds used in other studies.*

*Done.*

**RC2.11** Line 137: I think "1989-1960+1" was not intended to be included in text

*Yes, thank you for noticing this typo, since removed.*

*Done.*

**RC2.12** Lines 159-161: I think the connection between temperature and precipitation is an issue worth discussing. The wet years with poor yields also tend to be relatively cool (especially during foundation). The dry years tend to be hot.

*Thank you for the suggestion; we will discuss this point in the revised text. We also note how this fits well with your other queries about simultaneous changes or anomalies in temperature and precipitation.*

*Done.*

**RC2.13** Line 191: I don't see 1976 on the figure, and 2013 and 2018 don't seem particularly extreme.

*Good spot, thank you; we will remove 1976 and adjust this statement accordingly.*

*Done.*

**RC2.14** Line 200: This somewhat undercuts your preceding results. You do find climate-yield relationships so I don't see strong basis for claiming they are masked by inputs. Further, it is not clear which inputs these would be. I do not know of any short-term adaptive solutions to excess moisture (farmers can improve drainage and soil texture over time, but not within a single season). Further, the usual adaptive management for heat or drought is irrigation, which is not widespread in the UK. Instead, what might be more important/interesting is analyzing (or at least speculating on) the role of inputs in raising mean yields (over decades), and how that may influence yield variability (which you are trying to attribute differentially to climate).

*Thank you - this is a good point and we will adjust the text by editing the statement. (The reviewer is referring to the statement that "the relatively input-intensive nature of UK wheat production may be sufficient to mask crop responses to climatic variation"). Here we could replace "mask" with "dampen" (i.e. we still see effects by not as much as we might expect) and drop "inputs" (i.e. we refer to all management here, not just agrochemical inputs), i.e. "the relatively intensive nature of UK wheat production may be sufficient to dampen crop responses to climatic variation".*

*We would argue that there is still a role for agronomic management in dampening apparent relationships with climate - these might not be as direct as irrigating in response to drought, but farmers can, for example, change fungicide regimes to response to increased fungal disease brought about by wetter conditions. Farmers can also change many other aspects of management, including wheat variety, tillage, sowing date, sowing rate, or harvest date, in response to forecast or current conditions. We agree that these are not inputs as such and will change the wording.*

*We will also add some discussion of the role of inputs in raising mean yields and the current yield plateau (e.g. Knight et al. 2012).*

*Knight, S., Kightley, S., Bingham, I., Hoad, S., Lang, B., Philpott, H., Stobart, R., Thomas, J., Barnes, A., Ball, B. (2012) Desk study to evaluate contributory causes of the current yield plateau in wheat and oilseed rape. https://projectblue.blob.core.windows.net/ media/Default/Research%20Papers/Cereals%20and%20Oilseed/pr502-summary.pdf*

*In section 3.3 we have added further discussion of the role of agronomic management in dampening the relationships between crop yields and climate. We believe the text now provides a more rounded representation how yields might vary.*

**RC2.15** Line 205: This claim is interesting and usefully motivates the next section, but needs work, and here's one place using multivariate regression may be useful. In this more standard method, multiple climate variables together usually explains less than half of yield variation (full-model adjusted r2 < 0.5). Using individual pairwise correlations is less common, and so it's unclear what would be high or low correlation. If the correlations are indeed low in a more robust assessment, it could be because of the myriad other environmental or social factors contributing to yield (climate explains less than half of yield variability).

*Thank you. We will test the multivariate regression, as discussed in comment RC2.1, and include it if it proves logical to do so. These comments are helpful.*

*We developed a simple multiple regression model using the same variables that were initially included in the combined climate score (please see the new manuscript section 2.5 on "Statistical approach"). This regression model provided more refined results than the climate score and thus was retained in the revised manuscript. We then developed data-driven projections of future crop yields (forcing the same regression model with climate model outputs), which are shown in the revised Figure 6.*

**RC2.16** Line 257-259: Here's a place you could mention multivariate change. Cool and wet foundation phases have been linked to poor yields, and these are connected because it is hard to warm up the surface when soils are wet, and hard to dry out wet soils when it is cool. The projected warmer and wetter conditions are orthogonal to this connection, and some of that warming may help dry out waterlogged soils. Question is whether the warming will suffice to offset the increased precipitation, and this is the kind of question that a multivariate regression model could help answer. See for instance Rigden et al. 2020, Lesk et al. 2021, Ortiz-Bobea et al. 2019.

*Thank you for the suggestion and the references - we will include these citations and summarise their findings as points of discussion.*

*We have developed the discussion of multivariate climate change at different points in the text, including here; thank you for the helpful suggestions. We also employed the new data-based regression model to describe and discuss the interaction between different climate variables and their impacts on crop yield.*

**RC2.17** Line 265: Precipitation may not change much, but there is still warming, which will increase atmospheric vapour demand (all else being equal). So here's a place where some acknowledgement or analysis of multivariate change would probably lead to more robust conclusions about the future. Zampieri et al. 2017 touches on some of this multivariate influence. See also Lobell et al. 2013 with detail on the evaporative role of temperature (it's for U.S. maize, but relevant to interpreting future warming).

*Thank you for the helpful suggestions! We will discuss multivariate change here and consider using the model for future projections (as mentioned in our reply above to RC2.1). We will also include these suggested references.*

*Done.*

**RC2.18** Line 277-279: Yes, especially since temperature could have non-linear impacts, see Barlow et al. 2015, a useful reference for frost effects too.

*Thank you for the reference!*

*Included in section 3.5*

**RC2.19** Line 285-288: Great, this offsetting is coming to light as an important mechanism/uncertainty, I just think it could be discussed in more depth.

*Thank you! We will indeed discuss it in more depth in the revised manuscript.*

*We have featured the offsetting as a more prominent part of the manuscript results, thank you!*

**RC2.20** Line 300-301: Consider using term 'compound extremes' here and in the intro to link to emerging literature on this topic. E.g. Zscheischler et al. 2020.

*Thank you for the helpful suggestion. We will refer to compound extremes in both places.*

*Done.*

**RC2.21** Figures 4-5: I like that this shows the bivariate temperature-precipitation distributions. It is hard to differentiate the grey circles from diamonds, however. It may be easier to see if the 95% confidence ellipses are removed – I'm not sure what they add and could be replaced by simple dots showing point-estimates of mean yield. Otherwise, perhaps the climate model data should be presented on separate axes.

*Thank you for the suggestion. We will consider removing the ellipses and replacing them by simple dots, or alternatively using separate axes.*

*We decided to remove the ellipses and updated the figure accordingly.*

**RC2.22** Figure 6: this is a pretty convincing figure notwithstanding my concerns above, but it's hard to understand why the black data are showing y-axis values and an increasing trend, as I don't see yield projection results or methods anywhere in the paper. I assume the points are different years, and aggregate climate scores evolve over time. If so, this data should probably be separate time axes. The black data also seem visually like trendlines on the yield/climate score scatters, but I don't think they are so this may mislead readers.

*These comments are helpful - Figure 6 was evidently not clear enough. We will update this figure after having tested the multivariate model suggested in RC2.1. The idea of showing how the combined climate score might evolve as a time series is a particularly nice suggestion. This is a good way to merge contemporary data with the model projections.*

*We have removed the old Figure 6 showing the combined climate score and replaced it with a new time series figure that shows the future projected yields as a time series based on the multiple regression results (which provide more refined results than the former climate score).*

**References:**

Ainsworth, E. A., & Long, S. P. (2021). 30 years of freeâair carbon dioxide enrichment (FACE): What have we learned about future crop productivity and its potential for adaptation?. Global Change Biology, 27(1), 27-49.

Barlow, K. M., Christy, B. P., O'leary, G. J., Riffkin, P. A., & Nuttall, J. G. (2015). Simulating the impact of extreme heat and frost events on wheat crop production: A review. Field Crops Research, 171, 109-119.

Lesk, C., Coffel, E., & Horton, R. (2020). Net benefits to US soy and maize yields from intensifying hourly rainfall. Nature Climate Change, 10(9), 819-822.

Lesk, C., Coffel, E., Winter, J., Ray, D., Zscheischler, J., Seneviratne, S. I., & Horton, R. (2021). Stronger temperature–moisture couplings exacerbate the impact of climate warming on global crop yields. Nature food, 2(9), 683-691.

Lobell, D. B., Hammer, G. L., McLean, G., Messina, C., Roberts, M. J., & Schlenker, W. (2013). The critical role of extreme heat for maize production in the United States. Nature climate change, 3(5), 497-501.

Miralles, D. G., Gentine, P., Seneviratne, S. I., & Teuling, A. J. (2019). Land–atmospheric feedbacks during droughts and heatwaves: state of the science and current challenges. Annals of the New York Academy of Sciences, 1436(1), 19.

Ortiz-Bobea, A., Wang, H., Carrillo, C. M., & Ault, T. R. (2019). Unpacking the climatic drivers of US agricultural yields. Environmental Research Letters, 14(6), 064003.

Ortiz-Bobea, A., Ault, T. R., Carrillo, C. M., Chambers, R. G., & Lobell, D. B. (2021). Anthropogenic climate change has slowed global agricultural productivity growth. Nature Climate Change, 11(4), 306-312.

Ray, D. K., West, P. C., Clark, M., Gerber, J. S., Prishchepov, A. V., & Chatterjee, S. (2019). Climate change has likely already affected global food production. PloS one, 14(5), e0217148.

Rigden, A. J., Mueller, N. D., Holbrook, N. M., Pillai, N., & Huybers, P. (2020). Combined influence of soil moisture and atmospheric evaporative demand is important for accurately predicting US maize yields. Nature Food, 1(2), 127-133.

Zscheischler, J., Martius, O., Westra, S., Bevacqua, E., Raymond, C., Horton, R. M., ... & Vignotto, E. (2020). A typology of compound weather and climate events. Nature reviews earth & environment, 1(7), 333-347.

Citation: https://doi.org/10.5194/esd-2021-92-RC2

*All the recommended references have been added to the revised manuscript (carefully cited at the appropriate points). Thank you for the helpful review!*

**Response to Reviewer Comment 3 (RC3):**
'Comment on esd-2021-92', Anonymous Referee #3, 20 Jan 2022

The authors present an interesting analysis of UK's wheat yield variability. They first explore the influence of different climatic conditions on wheat yields to then construct a scoring system for combined climate effects. In their analysis, they separate major plant development stages. Finally, they use climate model projections to estimate potential yields in a warmer climate.

Despite the confined regional focus, I would expect that the findings and the presented approach would be of interest to a wide readership. The manuscript is well written.

*We thank the Reviewer for their positive assessment of our paper and we are glad they find that it is of interest to a wide readership. We describe how we propose to address each of their comments and suggestions, point-by-point, as listed in blue italic font below.*

Major concerns:

**RC3.1** The paper is based on statistical analysis and this analysis should be described in more detail including a description of underlying assumptions. Especially the part about the scoring system should be better introduced and potentially justified.

*We agree with this comment and will provide a complete description of the statistical methods and assumptions. We will also better introduce, describe and justify the scoring system. We will ensure that a reader can reproduce the methods in full.*

*This request mirrors that of the other two reviewers. Hence, we have included a new section describing the statistical approach in full (section 2.5 "Statistical approach"). This section also introduces, describes and justifies the new more standard multiple regression model which has now replaced the scoring system.*

**RC3.2** The analysis of climate effects during the plant development phases delivers interesting results. The authors argue that with their scoring system they can assess the combined effect of climatic conditions throughout the plant development. Here the question arises whether the climatic impacts during the production phase are the same irrespective of the climatic conditions throughout the earlier plant development stages. For example, Ben-Ari et al. 2018 describes a compound event where the combination of warm winter and wet spring lead to a crop failure. As I understand the analysis, it wouldn't be able to capture such compound events if it is not generally bad for wheat to have warm winters and wet springs. This is just an example, but it might help to understand a limitation that comes from splitting up events. I

would find it interested to read the authors view on this concern. These reflections could also be included in the discussion.

*The Reviewer raises an important point, which is that of compound "memory" effects across different plant development phases (e.g. a warm winter followed by a wet spring). We tried to capture the effect of different climatic conditions in different individual phases through our combined scoring system. However, it is correct that our approach does not capture the extent to which climatic impacts may depend on anterior climatic conditions (as this would require a different modelling approach). We will therefore add to the discussion this potential limitation of our approach, as suggested by the Reviewer. We will also try developing a multivariate regression approach as discussed in our reply to RC2.2, and will retain this approach if it provides more robust results.*

*We have included a discussion of this assumption/limitation (capturing the effects of antecedent climate conditions) in the manuscript new section on Assumptions and limitations (section 2.6). We have also developed a multiple regression model using the same variables that were initially included in the combined climate score. Since this regression model provides more interesting results than the previous approach (the combined climate score), we have kept instead the regression-based approach in the revised manuscript.*

**RC3.3** The use of only one climate model appears problematic to me. Furthermore, for this type of analysis I don't see the benefit of high spatial resolution if in the end regional averages are used. I would find it more convincing to see a CMIP6 ensemble instead of one high-resolution model. On the other hand, the climate model projections are not the main part of the analysis. Therefore one could also think of comparing this climate model to the CMIP6 ensemble and discussing the differences and potential biases.

*We appreciate the Reviewer's concern, however, the 12 UKCP ensemble members correspond to perturbed-physics experiments (PPE) of a single forcing Earth System Model (ESM), where uncertain parameters within the physics of the driving global model are varied. Thus the 12 members of the high resolution ensemble sample uncertainty in changes in the large-scale conditions due to modelling uncertainty and internal climate variability (so it is not really 'one model'). We acknowledge that the ensemble lacks information from other international climate models, and this is something that could be addressed in future work exploiting new CMIP5 downscaled UKCP Local projections that are underway within the UKCP project.*

*To answer the point about the benefits of high spatial resolution, the fine-scale information may still be relevant despite the spatial aggregation. This is because the high resolution model better captures the small-scale processes (in particular convection) behind extreme weather events, and this improved process representation can have an imprint on spatially aggregated fields. We appreciate that the Reviewer notes that the projections are just one part of the analysis, and therefore as suggested, we will compare this climate model in detail to the CMIP6 ensemble, and discuss the differences and potential biases within the revised manuscript.*

*This potential concern has also been raised by our other reviewers. Therefore, in the revised manuscript we more clearly explain that the high-resolution UKCP Local simulations are driven by a perturbed physics ensemble, and that they have wider uncertainty than is typically represented in one single climate model. We have added a new section titled "Assumptions and limitations" (section 2.6) which explains the use of the perturbed physics ensemble as described above.*

Minor comments:

**RC3.4** The abstract could be improved. At the moment it reads a bit like a summary of different results and ideas. The aim of the study should be clarified more precisely and not all results have to be included in the abstract.

*We will streamline the abstract to better highlight the aim of the study, the key outcomes, and implications, as suggested by the Reviewer.*

*We have carefully updated the abstract as suggested by the reviewer.*

**RC3.5** L9-10: "future impacts of climate projections on wheat". I think this should be formulated differently.

*Thank you for spotting this. We will rephrase the sentence so that it addresses the impacts of possible future changes in climate on crop yields.*

*Done.*

**RC3.6** L30-31: Is this due to climatic conditions only? Or does technology play a role here?

*The Reviewer is referring to the statement that "the UK climate has historically been well suited to growing wheat". Technology and investment in the agricultural sector have certainly played their part in the current wheat yields seen in the UK (as can be seen from the increasing trend in Figure 1a as technological and agronomic innovations were introduced). However, wheat is a temperate species, and the UK climate is particularly well suited to its development when autumn-sown. For example, Harkness et al (2020) state:*

*"As a temperate species the typical weather conditions of western Europe, including the UK, are favourable for wheat production (Reynolds et al., 2010). Approximately 40% (~1.8 million hectares) of the arable cropping area in the UK is dedicated to wheat production (DEFRA, 2018). Despite the relatively small acreage, the UK produces approximately 2% of the world's wheat benefitting from a high average yield of ~8 t ha−1, compared to a world average of ~3.5 t ha−1 (FAOSTAT, 2018)"*

*We will add similar wording and appropriate citations to clarify this.*

*Done.*

**RC3.7** L97-101: Did you consider a different spatial aggregation methods for precipitation? While for temperature it seems reasonable to average over the regions, for precipitation there could be other meaningful choices. As an example, what would you think about area affected by extreme precipitation instead of regionally averaged precipitation?

*We did consider other approaches, such as the highest rainfall event within each region, rather than the regional average. Overall, we found the regional average produced more meaningful results. We did not consider the fractional area affected by extreme precipitation, but we accept this is an approach worth testing in future work. Hence we can mention this additional potential statistic in the limitations/further work section of the revised manuscript.*

*We discussed this point in the new "Assumptions and limitations" section of the manuscript (section 2.6).*

**RC3.8** L111: I think you should mention here, that the scientific community is not considering this scenario as a plausible future. I have seen, that you do so later on. Maybe still worth mentioning earlier.

*We will mention this point earlier in the manuscript. We are constrained by the single scenario of the UKCP Local simulations (which is RCP8.5 of course). RCP8.5 was deliberately chosen as the configuration for UKCP Local simulations to maximise the signal to noise. Using a high*

*emissions scenario has the advantage that we can infer changes for other lower emissions scenarios using scaling approaches.*

*We included a discussion of this point in the new "Assumptions and limitations" section of the manuscript (section 2.6).*

**RC3.9** L102: Although the UKCP Local simulations are surely great, there remains a large uncertainty with respect to forced changes in precipitation. The accurate representation of small features in these simulations does not necessarily reduce the uncertainty concerning the regional trend in precipitation. Therefore it would be good to compare the precipitation tendency from this model with climate models from other institutes. I have seen that you do so later in the manuscript.

*The Reviewer makes an important point, and we agree. We will make sure that the trends from the UKCP Local model are explicitly compared with those from climate models from other institutes. As mentioned above in response to RC3.3, the driving ESM of UKCP Local is subjected to perturbed physics, so is intended to at least partially represent the range of uncertainty in climate models. Therefore, the trends of the UKCP Local simulations should at least partially cover the range of trends of the ESMs (see last bullet point of page 5 of this document: https://www.metoffice.gov.uk/binaries/content/assets/metofficegovuk/pdf/ research/ukcp/ukcp18-factsheet-local-2.2km.pdf).*

*We have added a discussion of the differences between the UKCP Local simulations and other CMIP models in the new "Assumptions and limitations" section of the manuscript (section 2.6). We also note that although we have sympathy with the reviewer regarding what other models may eventually project, as yet there are only a very small number of research centres that have produced projections at storm-resolving resolutions of just ~ two kilometres.*

**RC3.10** L120: Could you add one or two sentences on the bias correction method? Is it a trend-preserving bias correction?

*Yes, the bias correction preserves any trend. It is a simple scaling method, which is additive for temperature and multiplicative for precipitation (therefore it preserves an absolute or relative trend, respectively). We will describe the method in more detail in the revised text and make sure this is clear.*

*Done.*

**RC3.11** L173-175: Are these two sentences contradicting each other?

*The Reviewer is referring to "While crops are growing rapidly during the Construction phase (April to early June), both late frosts and dry weather can reduce crop growth (Table 1). For this period in each year, we find no significant associations between climate characteristics and crop yields (Table 2)." This is not necessarily a contradiction, as reduced growth does not always carry through to reduced yield. We will clarify this point in the revised manuscript.*

*Done.*

**RC3.12** Section 3.2 and Table 2: How would you explain that the effects of climate conditions are different between the regions? I wouldn't have expected different effects for the different regions. If there is a reason for that it would be good to mention it. You explain this in L194-206, right?

*Yes. The climate conditions and UKCP Local climate projections are not universally identical across all the UK. For example, rainfall tends to be more frontal in the north (with orographic rainfall over high ground), and more convective in the southeastern UK. The climate*

*projections also exhibit gradients in the changes across the UK. Even in a single ensemble, there are north-south gradients in the future changes in rainfall which can be quite different to the present-day climatology and relate to regional differences in increases in moisture availability as well as changes in circulation patterns.*

*Additionally, the association between climate anomalies and wheat yields can be explained by combinations of i) resilience of the wheat plant; ii) husbandry practices of farmers and agronomists (lines 194-206); and iii) non-climatic biophysical conditions (e.g. soils, day length), all of which may vary regionally.*

*In the revised manuscript, we will more clearly explain how these factors vary regionally, and we will provide further explanation of how the climate projections might affect the crop yields in light of these regional differences.*

*We have added some sentences in section 3.2 describing gradients in climatic and biophysical conditions and their effects across the regions.*

**RC3.13** L220: What is the advantage of using this "score". Couldn't you also work directly with the correlations of table 2?

*The idea behind the score is that if climate conditions are very poor or very good in just one of the crop growth stages then the effects may not be sufficient to alter crop yields. This is because there are multiple factors which affect crop growth. For instance, poor conditions in one stage may be mitigated by good conditions or agronomic methods in another stage (e.g. wet weather leading to increased incidence of fungal disease can be mitigated by subsequent increased use of fungicides). In contrast, detrimental climate conditions may have a cumulative impact across multiple growth stages, and this would be reflected by our score statistic. We will ensure this point is clarified in the revised manuscript.*

*We have removed the combined climate score because the new multiple regression approach provides clearer results and projections of future yields under climate change, and is easier to interpret. We have explained the methods in detail in the new section 2.5, "Statistical approach".*

**RC3.14** L246: Is "sample" the correct word here? I would have written "project". But I'm not a native speaker.

*The Reviewer is referring to "UKCP simulations tend to sample greater future warming and drying in summer compared to the full CMIP5 ensemble". We will modify the wording to make it more explicit (e.g. "tend to project").*

*We have removed the use of the term "sample" in some places and used verbs which are more intuitive (notably "project" and "describe"). Elsewhere in the manuscript, when "sample" is the most accurate term, we have retained it.*

**RC3.15** L246: Is this statement true for the UK in particular? And how did you get there? I think it would be good to spend a few more sentences on this aspect to provide a good overview of potential biases over UK.

*Yes, the statement that "UKCP simulations tend to sample greater future warming and drying in summer compared to the full CMIP5 ensemble" is true for the UK.*

*Our response to RC1.22 is relevant here, and we will add a more detailed discussion in the manuscript of how the UKCP ensemble compares to the CMIP ensemble. The figure in RC1.22 from Kendon et al. (2021) shows that UKCP Local projections are generally within the 5-95% probability levels of the UKCP Probabilistic projections (which include some multi-model information from CMIP5). One exception is winter when the UKCP Local show some*

*precipitation responses above the 95% level. This is understood and relates to the improved representation of winter-time convective showers in the Local 2.2km model (Kendon et al 2020) used in our study. UKCP Local projections sample relatively high temperature changes. Changes in summer precipitation show a considerable drying in the Local projections (2.2km), whereas the 13 CMIP5 simulations and the UKCP Probabilistic projections indicate that outcomes with more modest reductions or small increases should also be considered.*

*References*

*Kendon E J et al (2021) Update to UKCP Local (2.2km) projections, July 2021, Met Office, Exeter, UK https://www.metoffice.gov.uk/pub/data/weather/uk/ukcp18/science-reports/ukcp18_local_update_report_2021.pdf*

*Kendon, E. J., et al (2020) Greater future UK winter precipitation increase in new convection-permitting scenarios. J Climate. DOI: 10.1175/JCLI-D-20-0089.1*

*The manuscript has been updated by explaining the potential biases over the UK: a new section has been added (2.6 Assumptions and limitations) which describes how the UKCP Local projections compare with other climate model projections.*

**RC3.16** Figure 7: I think this figure could be improved a bit. What do you think about displaying the ensemble spread by a shaded area and the ensemble median by a line?

*We agree the figure can be improved by displaying the ensemble spread as a shaded area and the ensemble median by a line. This will simplify and hopefully clarify the figure in the revised manuscript. We appreciate this suggestion.*

*Done.*

**RC3.17** L292: "since crop yields" instead of "since inter-annual crop yields"?

*Yes, we will change this text accordingly.*

*Done.*

*Thank you for the helpful review!*

Citation: https://doi.org/10.5194/esd-2021-92-RC3

---

## Referee Report (RR1)

Thanks for your efforts revising the paper. I think the revisions add some needed caveats and statistical grounding to the conclusions. I just have a few minor clarifications to suggest:

1) The word 'escape' which now appears first in abstract

It's not clear what this means in the abstract, so this should probably be clarified a bit.

Also, I'm not sure this it is a well supported conclusion that the effects you are showing are indeed due to bad weather in multiple growth stages exceeding the ability of farmers to adapt. It could be purely due to plant physiology, and you don't show data on farmer management responses in response to weather. This point on 'escape' seems like more of a speculative discussion point, so maybe try to frame it more as such.

A related point is that, on line 349, it's not clear how this inter-stage compensation conclusion is drawn.

One piece of statistical evidence you could note on this front is the improved r2 in the multiple regression compared to the pairwise correlations. e.g. nationally, pearson r is about 0.46 at most, equivalent to r2 of 0.21, but the model adjusted r2 is about 0.3 nationally (~10% more variance explained, or a 50% improvement over the pairwise correlations). In other words, the model is capturing something individual variable correlations do not (and this could be that compensation). That said, whether this added explanatory power is from inter-stage compensation, or compensation between variables within a singe stage, is not clear from the regression results alone. That should at a minimum be acknowledged as a limit on your conclusions, and something worth following up on (could fit well around line 455). Further, it's adds to my questioning your prominent conclusion about multi-stage weather anomalies 'escaping' farmer adaptive capacity.

2) Variable selection for regression

I more or less buy this variable selection, but most studies using these methods test a variety of potential models and variable combinations. So simply stating on Line 178-179 that "Alternative metrics could also be selected, such as var_dailyT or var_maxT in the Production phase, or days_P>10mm in either phase, but these are likely to show similar relationships" makes me a bit uneasy. Especially since you talk a lot about change in precipitation extremes in your projections. So maybe simply mention somewhere that future research should further develop this statistical crop model.

3) A few small things.

Line 209: this is rather vague: "meaningful results" how?

Table 4: would be helpful to see significance (SE or p-value) of coefficients (can use star scheme as in other tables), and units of coefficients (I think t/ha for intercepts, t/ha/°C or t/ha/mm for slopes)

Line 380: good point, maybe worth mentioning that this interaction depends on how the link between precipitation and soil moisture change in the future (a topic drawing increasing attention both in climate and crop science, enabled by the rise of satellite-derived soil moisture observations).

Line 382: contrary to what expectations? Actually, your projections make sense given UK wheat's climate sensitivity. Maybe clarify that you mean contrary to global expectations of declining yields under climate change. Yield gains due to reduced frost risk in cool climates are widely expected. The drying effect of warming temperatures in places where often crops get hit by sodden conditions is perhaps less widely appreciated.

A final thing occurred to me reading the revised paper, which is the irony or injustice that the country that initiated the rise of fossil fuels (and gained wealth and adaptive capacity doing so) ultimately ends up benefitting from the consequences of climate change (at least in terms of agriculture). If you want to note the wider implications of your study, you could note this point in the conclusions (and its implications for the UK's ethical obligations to finance adaptation in places that did not get wealthy off of fossil fuels, but where climate change will lower yields).

---

## Author Response (AR2)

**Resilience of UK crop yields to compound climate change**

**Responses to reviewer comments**

*August 2022*

**Editor decision: Publish subject to minor revisions (review by editor)**

21 Jul 2022
by Gabriele Messori
Comments to the author:
Dear Authors,

While all Reviewers recognise the improvements in your revised text, Reviewer #1 still raises some concerns on the appropriateness of the statistical analysis. Ensuring this analysis is sound is key to supporting the conclusions drawn in the paper.
I am returning the paper to you for minor revisions, but view a detailed and convincing reply to Reviewer #1 as essential before the paper may be accepted for publication. Please contact the editorial office should you need more time than that automatically allocated by the editorial system.

Best Regards,
Gabriele Messori

*We would like to thank the Editor for his evaluation of our paper. We also thank the Reviewers for their additional comments, which have helped further improve the manuscript. Please find below a detailed reply to all Reviewers, describing how we have addressed their comments.*

**Report 1 (Anonymous referee #3)**

In the revised manuscript, the description of statistical methods is greatly improved and a section on assumptions and limitations helps the interpretation of the results and addresses many concerns raised in the reviews. The added section on the statistical approach as well as the section on assumptions and limitations are a great improvement of the manuscript. The concern of reviewer 1 about the lack of statistically significant correlations between climate variables and yields however remains and the newly added multi linear regression does not dispel this concern.

*We are glad the Reviewer finds the manuscript greatly improved, and are grateful for the additional comments. Regarding the correlations between climate variables and yields, we agree with the importance of stating which associations are statistically significant and which are not. To this end, we have added sentences explicitly emphasising which correlations are significant and which are not, in our revised manuscript (as described in the next point).*

One aim of the study is to investigate "(1) whether statistically significant associations exist between observed temperature/precipitation metrics and historical wheat yields" (L73-74). My reading of table 2 would be that such statistical associations only exist for the production

phase and for region EMYH and NAT. In the conclusions it should be declared that for the two other phases and for the other regions there is no statistical association.

*We agree and further elaborate this point in the Conclusion. Weakly significant correlations are found also for some variables in SEE (min_meanT and min_minT) and SNE (var_dailyT), so we write:*

> *"Significant statistical associations are found principally in the Foundation and Production phases and principally for regions EMYH and NAT." (see Conclusions)*

Furthermore, the authors do not provide sufficient evidence for the trustworthiness of the multi linear regression model and for many interpretations drawn from it. As an example: From table 4 it appears that the multi linear regression model for the region SFE has a p-value of below 0.05 (I assume that this is the p-value of some goodness of fit test but it would be good to clarify this). However, from table 2 we know that none of the used climate variables is significantly correlated to yields. Could it be that this low p-value for the model is a result of over-fitting? A 5 parameter model fitted to roughly 30 years of data might be a bit much especially if none of the input variables is significantly correlated to the fitted variable (yields).
One way of excluding the risk of over-fitting would be to separate training testing and validation data, but this would require even more data. In my view such a model could be applied to the national yield anomalies as displayed in Figure 1c with detrended estimates of temperature and precipitation if the yield data and climate data before 1980 can be trusted.

*The p-value is that of the multiple linear regression model; specified in the Table caption. The variables (max_minT and total_P in the Foundation phase and max_maxT and total_P in the Production phase) are chosen because of the consistent sign of the associations found with yields across regions and nationally (shown in Table 2). The strength of the association does vary regionally, as discussed previously.*

*The Reviewer makes a valuable point regarding the reliability of a multiple linear regression model trained with only a few decades' worth of available data. To address the Reviewer's suggestion, we computed the "Predicted R2", a metric which evaluates the ability of a linear model to predict responses for new observations (this metric removes a data point from the dataset, calculates the regression equation, evaluates how well the model predicts the missing observation, and repeats this process for all data points in the dataset) for the national model and find that it is 0.091, which suggests there is some predictability, although limited over the 31-year period, using this metric. In the revised manuscript, we discuss the low predictability and emphasise that our model is a proof-of-concept that could be refined once longer data becomes available in the future. Section 2.5 now includes the following statement:*

> *"Although the model is significant (p<0.05) in EMYH, SEE and NAT, the predictability is relatively low. Alternative metrics could also be selected to improve the model, such as var_dailyT or var_maxT in the Production phase, or days_P>10mm in either phase, or additional variables reflecting e.g. precipitation intensity. These variables have not been tested and should be evaluated in future research, further developing the statistical crop model. Our model is a proof-of-concept that could be refined to improve the predictive skill if further data becomes available."*

*Additionally, as highlighted by Reviewer 2, the multiple regression has an improved R2 compared to the pairwise correlations. We have thus included the following text as recommended by Reviewer 2:*

*".. the strongest associations between climate and yield anomalies may occur during years with cumulative climate impacts across growth stages. Cumulative impacts can be seen in the improved R2 in the multiple regression compared to the pairwise correlations. In other words, the model is capturing something individual variable correlations do not, and this could be that compensation across phases. That said, whether this added explanatory power is from inter-stage compensation, or compensation between variables within a single stage, is not clear from the regression results alone."*

*Table 4 has also been revised as suggested by Reviewer 2.*

My suggestion would be to include a paragraph in the beginning of section 3.3 that states that most of the correlations are not statistically significant. Than one could explain potential reasons for this, including the "combined resilience of the wheat plant (i.e. physiological reproductive mechanisms) and the husbandry skills of farmers and agronomists" but also the length of the observational record. The interpretation of the lacking statistically significance should be that based on the time series of one region one can not exclude that the correlation occurs by chance. As the authors already suggest by marking some parts in table 2 in bold, the fact that correlation coefficients go in the same direction in all three regions could however be an indication for a link between the variables. In combination with a plausible explanation this link can be interpreted, but should not be over-interpreted.

*We agree and have updated the text in section 3.3 accordingly:*

*"Most of the correlations in the historical data are not statistically significant (Table 2). The often relatively weak association between climate anomalies and wheat yields at the level of individual growth stages can be explained partly by the shortness of the observational record, the combined resilience of the wheat plant (i.e. physiological reproductive mechanisms) and the husbandry skills of farmers and agronomists in mitigating these impacts by adjusting to climatic extremes"*

In that case, the prior knowledge on the mechanisms that lead from climate variables to yields plays an important role. Without this prior knowledge one could not exclude that the discussed associations between climate variables and yields might just be random. I would therefore suggest to include this prior knowledge in a structured way in the introduction instead of introducing it throughout the results section.

*We agree with the Reviewer that prior knowledge plays an important role in the choice of climate variables and their interpretation, in relation to crop yields. This prior knowledge is summarised succinctly in Table 1, and is highlighted throughout the manuscript. We add an additional sentence when introducing Table 1, which states:*

*"Prior knowledge on the effects of climate in different growth stages guides our choice of climate variables in the study (Table 1)."*

I also have a question about your interpretation of figure 4 and figure 5: Do these figures really "reveal the important connection between temperature and precipitation" as you state in line 273 or do they mostly show that precipitation and temperature are not independent from each other. For instance, it is to be expected that in the mid-latitudes wet summer periods are cooler than dry summer periods. This does not necessarily imply that for wheat growth both temperature and precipitation are important.

*We have modified the statement, which now reads:*

> *"The figures also indicate that temperature and precipitation are not independent from one another"*

Minor comments:

L174-178: I would rather write the climate indicators in words than in formula notation. Without table 2 this is difficult to read.

*The climate indicators are explained above in section 2.2 and are very long, which we feel would make the paragraph even harder to read. Therefore, we prefer to leave the formula notation here.*

L219-220: I would drop the "RCP8.5 is still a plausible scenario" as it is not fully correct. Furthermore, the next sentence gives the justification for the use of the RCP8.5 scenario.

*Agreed. The text has been updated as:*

> *While the likelihood of such high on-going emissions is now considered low (Chen, D. et al., 2021; Hausfather and Peters, 2020), the RCP8.5 scenario is commonly used to facilitate detection of climate signals in future projections above natural variations in the climate (due to the large changes projected), and was deliberately chosen as the configuration for UKCP Local simulations to maximise the signal to noise.*

L230-236: Where did you get this infomration from CMIP5 from? This should be referenced somewhere. I just checked here: https://interactive-atlas.ipcc.ch/regional-information

*It is not entirely clear which information the Reviewer is referring to as a reference is provided in the text (Kendon et al. 2020). However, we think this URL could be helpful for the readers and have added it in the manuscript, which now reads:*

> *"UKCP Local projections also project relatively high temperature changes compared with other climate models (see e.g. https://interactive-atlas.ipcc.ch/regional-information)."*

L234-236: This observation appears to be true for CMIP6 as well. And I think this is really important for the further interpretation of the projected future climate conditions. I think this observation should be acknowledged and discussed in the results section (3.4 or 3.5) conclusions as well.

*The sentence has been updated as:*

> *"Changes in summer precipitation show a considerable drying in the UKCP Local projections, whereas the CMIP5 and CMIP6 simulations indicate that outcomes with more modest reductions or small increases should also be considered."*

*We have updated the results section (in the paragraph that discusses future projections):*

*"It is important to note that the UKCP Local projects stronger drying than CMIP5-6 models."*

*And also the Conclusions:*

*"the projected significant decreases in future rainfall (which are stronger in UKCP Local than in CMIP5 and 6) could equally be beneficial to wheat yields"*

L374-378: This list of previous research does not fit to well in the results section in my opinion. To me this reads more as a discussion that helps to put your results into perspective and should therefore not come before your results.

*We agree these sentences fit better elsewhere and have moved them to the second paragraph of the introduction.*

*We greatly thank the Reviewer for their help in improving the manuscript!*

**Report 2 (Referee #2: Corey Lesk)**

Thanks for your efforts revising the paper. I think the revisions add some needed caveats and statistical grounding to the conclusions. I just have a few minor clarifications to suggest:

*We would like to thank Corey Lesk for his additional helpful comments on our manuscript.*

1) The word 'escape' which now appears first in abstract
It's not clear what this means in the abstract, so this should probably be clarified a bit. Also, I'm not sure this it is a well supported conclusion that the effects you are showing are indeed due to bad weather in multiple growth stages exceeding the ability of farmers to adapt. It could be purely due to plant physiology, and you don't show data on farmer management responses in response to weather. This point on 'escape' seems like more of a speculative discussion point, so maybe try to frame it more as such.

*This comment about "escaping" farmers' ability to adapt has now been removed from the abstract and the short summary to avoid any speculation.*

A related point is that, on line 349, it's not clear how this inter-stage compensation conclusion is drawn.
One piece of statistical evidence you could note on this front is the improved r2 in the multiple regression compared to the pairwise correlations. e.g. nationally, pearson r is about 0.46 at most, equivalent to r2 of 0.21, but the model adjusted r2 is about 0.3 nationally (~10% more variance explained, or a 50% improvement over the pairwise correlations). In other words, the model is capturing something individual variable correlations do not (and this could be that compensation). That said, whether this added explanatory power is from inter-stage compensation, or compensation between variables within a singe stage, is not clear from the regression results alone. That should at a minimum be acknowledged as a limit on your conclusions, and something worth following up on (could fit well around line 455). Further, it's adds to my questioning your prominent conclusion about multi-stage weather anomalies 'escaping' farmer adaptive capacity.

*We have modified the statement on line 349 by including the Reviewer's suggestions:*

> *"Thus, the strongest associations between climate and yield anomalies may occur during years with cumulative climate impacts across growth stages. Cumulative impacts can be seen in the improved R2 in the multiple regression compared to the pairwise correlations. In other words, the model is capturing something individual variable correlations do not, and this could be that compensation across phases. That said, whether this added explanatory power is from inter-stage compensation, or compensation between variables within a single stage, is not clear from the regression results alone."*

*Additionally, in the conclusions, we acknowledge the limitation, also following the suggestions above:*

> *"However, it is unclear whether the added explanatory power of the regression model is from inter-stage compensation, or compensation between variables within a single growth stage. This would be an area for further research. This data-driven regression approach could additionally be refined by including various thresholds …"*

2) Variable selection for regression

I more or less buy this variable selection, but most studies using these methods test a variety of potential models and variable combinations. So simply stating on Line 178-179 that "Alternative metrics could also be selected, such as var_dailyT or var_maxT in the Production phase, or days_P>10mm in either phase, but these are likely to show similar relationships" makes me a bit uneasy. Especially since you talk a lot about change in precipitation extremes in your projections. So maybe simply mention somewhere that future research should further develop this statistical crop model.

*Thank you. We agree with this point. The sentence has been revised to:*

> *"Alternative metrics could also be selected, such as var_dailyT or var_maxT in the Production phase, or days_P>10mm in either phase, or additional variables reflecting e.g. precipitation intensity. These variables have not been tested and should be evaluated in future research, further developing the statistical crop model."*

3) A few small things.

Line 209: this is rather vague: "meaningful results" how?

*By this we meant stronger statistical associations. However, this point does not add value to the manuscript and so for the sake of clarity, these sentences have been removed.*

Table 4: would be helpful to see significance (SE or p-value) of coefficients (can use star scheme as in other tables), and units of coefficients (I think t/ha for intercepts, t/ha/°C or t/ha/mm for slopes)

*We have added the significance of the coefficients and their units.*

Line 380: good point, maybe worth mentioning that this interaction depends on how the link between precipitation and soil moisture change in the future (a topic drawing increasing

attention both in climate and crop science, enabled by the rise of satellite-derived soil moisture observations).

*Thank you. We have added this point in the revised text (it is now in the introduction, as Reviewer 1 felt that it didn't fit well in the Results section).*

Line 382: contrary to what expectations? Actually, your projections make sense given UK wheat's climate sensitivity. Maybe clarify that you mean contrary to global expectations of declining yields under climate change. Yield gains due to reduced frost risk in cool climates are widely expected. The drying effect of warming temperatures in places where often crops get hit by sodden conditions is perhaps less widely appreciated.

*Thank you. We agree and the text has been updated accordingly ("Contrary to global expectations of declining yields under climate change").*

A final thing occurred to me reading the revised paper, which is the irony or injustice that the country that initiated the rise of fossil fuels (and gained wealth and adaptive capacity doing so) ultimately ends up benefitting from the consequences of climate change (at least in terms of agriculture). If you want to note the wider implications of your study, you could note this point in the conclusions (and its implications for the UK's ethical obligations to finance adaptation in places that did not get wealthy off of fossil fuels, but where climate change will lower yields).

*Some of the authors of this paper do have sympathy towards this view. However, we do have to be broadly independent of statements about international affairs. There is also an issue of impartiality for some authors who work in a government-owned setting. We accept that this is an interesting point, but we request to not discuss it in the revised manuscript, to avoid any suggestion of presenting opinions.*

**Report 3 (Anonymous referee #1)**

I would like to congratulate the authors on an analysis well done and manuscript well written. Thank you for taking serious my concerns and confusion in places. In particular I'd like to mention the new section on assumptions, which is great both in terms of clarity and completeness.

*We are thankful to the Reviewer for their positive evaluation of our manuscript and their previous comments, which greatly improved the work!*

---

## Author Response (AR3)

**Resilience of UK crop yields to compound climate change**

**Responses to Editor's comments**

*September 2022*

04 Sep 2022
**Editor decision: Publish subject to technical corrections**

**Comments to the author**:
Dear Authors,

I believe that you have satisfactorily addressed the remaining reviewer concerns, and am glad to accept your manuscript for publication in ESD subject to techincal corrections. Specifically, i would suggest the following two edits:
- Mention the "predicted R2" somewhere in the text, else the statement added in Sect. 2.5 risks remaining somewhat cryptical to the reader. Moreover, this is a key point to addressing the overfitting concern raised by one of the reviewers.
- ll. 245-246 (in the tracked changes manuscript) Make sure that an appropriate reference is provided for this statement on the (CMIP5 and) CMIP6 models, as well as elsewhere where this point is raised.

Best Regards,
Gabriele Messori

*Dear Gabriele Messori,*

*Thank you very much for your acceptance and positive evaluation of the revised manuscript. We have added the Predicted R2 at line 186, and added a reference at line 247 of the revised manuscript, as requested.*

*With best regards,*
*Louise Slater*